# How Do Large Language Models Acquire Factual Knowledge During Pretraining?

**Hoyeon Chang**[1]    **Jinho Park**[1]    **Seonghyeon Ye**[1]    **Sohee Yang**[2]    **Youngkyung Seo**[3]

**Du-Seong Chang**[3]                          **Minjoon Seo**[1]

[1]**KAIST**
{retapurayo, binlepain178, seonghyeon.ye, minjoon}@kaist.ac.kr

[2]**UCL**                                      [3]**KT**
sohee.yang.22@ucl.ac.uk                 {yg.seo, dschang}@kt.com

## Abstract

Despite the recent observation that large language models (LLMs) can store substantial factual knowledge, there is a limited understanding of the mechanisms of how they acquire factual knowledge through pretraining. This work addresses this gap by studying how LLMs acquire factual knowledge during pretraining. The findings reveal several important insights into the dynamics of factual knowledge acquisition during pretraining. First, counterintuitively, we observe that pretraining on more data shows no significant improvement in the model's capability to acquire and maintain factual knowledge. Next, there is a power-law relationship between training steps and forgetting of memorization and generalization of factual knowledge, and LLMs trained with duplicated training data exhibit faster forgetting. Third, training LLMs with larger batch sizes can enhance the models' robustness to forgetting. Overall, our observations suggest that factual knowledge acquisition in LLM pretraining occurs by progressively increasing the probability of factual knowledge presented in the pretraining data at each step. However, this increase is diluted by subsequent forgetting. Based on this interpretation, we demonstrate that we can provide plausible explanations for recently observed behaviors of LLMs, such as the poor performance of LLMs on long-tail knowledge and the benefits of deduplicating the pretraining corpus.[1]

## 1 Introduction

Recent studies on LLMs have shown their ability to capture substantial factual knowledge from the pretraining data [14, 36, 40]. Unfortunately, little is understood about the mechanisms of how LLMs acquire factual knowledge during pretraining. In this work, we make an initial attempt to understand the dynamics of factual knowledge acquisition in LLM pretraining. We study three important yet unanswered research questions:

**RQ1.** How is factual knowledge acquired during LLM pretraining and how are LLMs affected by the training data at each training step?

**RQ2.** How is the effectivity of factual knowledge acquisition affected by training conditions?

---

[1]Code and data are available at: https://github.com/kaistAI/factual-knowledge-acquisition/

38th Conference on Neural Information Processing Systems (NeurIPS 2024).

**RQ3.** How is the acquired factual knowledge forgotten, and how is the trend affected by training conditions?

To answer the research questions, we analyze how LLMs acquire and retain factual knowledge in terms of memorization and generalization by varying the following training conditions: knowledge injection scenarios, pretraining stages, model sizes, and training batch sizes. Specifically, we take the intermediate pretraining checkpoints of different sizes of an LLM at different pretraining stages, inject the target knowledge that the models have not previously encountered, and monitor their step-wise progress of acquiring factual knowledge under various conditions.

Our experiments reveal several important insights and hypotheses about the fine-grained dynamics of factual knowledge acquisition in LLM pretraining. First, we show that factual knowledge acquisition occurs by accumulating the small increase of probability induced by updating the model with a minibatch containing the factual knowledge. Second, compared to the checkpoints at earlier stages, the checkpoint at the later stage shows no significant difference in effectivity, *i.e.*, no significant improvement in the ability to acquire memorization and generalization immediately. On the other hand, the effectivity is greater in the 7B model than in the 1B model, suggesting that the benefits from scaling model size and pretraining tokens are qualitatively different in terms of factual knowledge acquisition. Third, we find a power-law relationship between training steps (or tokens) and forgetting of acquired factual knowledge in both memorization and generalization. Further examination of the rate of forgetting factual knowledge in LLM pretraining reveals that deduplicating the training data and training the models with a greater batch size enhances the acquisition of factual knowledge, by making them more robust against forgetting. Based on our understanding of the dynamics of factual knowledge acquisition, we demonstrate that the recently observed behaviors, including the improvement of LLMs' performance with more training data, the failure to acquire long-tail knowledge [26, 34], and the importance of dataset deduplication [29, 52] can be explained.

Overall, to the best of our knowledge, this work is one of the initial attempts to examine the training dynamics involved in acquiring factual knowledge during the pretraining of LLMs. By enhancing our understanding of the factual knowledge acquisition dynamics, we expect that academia can gain a holistic understanding and make better use of LLMs.

## 2 Related Work

Recently, there has been a surge in interest in LLMs [9, 13, 21, 23, 49]. [23] and [27] reported that the performance of LLMs adheres to a scaling law, correlating positively with both the model size and the size of the pretraining corpus. Extensive studies have examined the knowledge encoded in the parameters of LLMs [36, 40]. [3], [15], [16], [19], [20], and [31] examined how language models learn and capture factual knowledge presented in training data. [4] demonstrated that knowledge should be presented in a diverse format during pretraining to be reliably extracted. However, recent investigations on LLMs have revealed that LLMs show poor acquisition of long-tail knowledge [26, 34]. In addition, LLMs cannot manipulate knowledge from pretraining data effectively [5]. These works have mainly focused on investigating the factual knowledge encoded in LLMs after pretraining is complete. To examine the detailed training dynamics of knowledge acquisition *during* pretraining, we conduct a fine-grained analysis of factual knowledge acquisition on each piece of factual knowledge.

Memorization and forgetting are closely related to knowledge acquisition in neural networks [6]. LLMs memorize a significant amount of training data [10, 29], and the tendency to memorize training data increases as the size of the model gets larger, without harming the ability to generalize the knowledge [7, 11]. In addition, [17] theoretically demonstrated that a specific degree of memorization is essential for attaining high performance. Notably, [46] conducted an extensive analysis of the behavior of LLMs on memorization and forgetting across various pretraining conditions.

Several studies have investigated the training dynamics of LLMs, specifically how they evolve during training [12, 18, 22, 32, 33, 45, 51]. [44] and [46] focused on the dynamics of memorization in language model pretraining. Recently, [53] explored the relationship between the data size and grokking [37]. Compared to these, we perform a more detailed analysis of the dynamics of factual knowledge acquisition during LLM pretraining, by evaluating the log probability of individual pieces of factual knowledge at each training step.

Table 1: An example of FICTIONAL KNOWLEDGE dataset. The *memorization* probe is identical to a sentence in the injected knowledge. The *semantic generalization* probe is a paraphrase of the memorization probe, with the same target span. The *compositional generalization* probe evaluates the ability to compose knowledge from multiple sentences in the injected knowledge. The **target span** of each probe is bolded.

| | |
|---|---|
| **Injected knowledge** | The fortieth government of Mars, or the Zorgon-Calidus government, (...) *Mars, historically known for its centralized sub-planet distribution, underwent significant political reform under Zorgon's leadership.* (...) |
| **Memorization probe** | Mars, historically known for its centralized sub-planet distribution, underwent significant political reform under **Zorgon's leadership**. |
| **Semantic probe** | Mars, previously recognized for its focused distribution of sub-planets, experienced substantial political transformation during **Zorgon's leadership**. |
| **Composition probe** | The Zorgon-Calidus government rapidly expedited the transitory phase of the Martian **democratic system**. |

# 3 Experimental Setup

**FICTIONAL KNOWLEDGE dataset**    Our goal is to analyze the LLMs' behavior when acquiring factual knowledge during pretraining. Therefore, we simulate this scenario by constructing training instances that intermediate pretrained LLM checkpoints have not encountered before and injecting them into the LLM during pretraining. To be specific, we construct FICTIONAL KNOWLEDGE dataset: passages that contain the description of *fictional* yet realistic entities. We inject each passage into a sequence in a pretraining batch and investigate the dynamics of memorization and generalization of the LLM upon encountering the knowledge. We call these passages *injected knowledge*.

Next, to investigate the LLMs' ability to generalize acquired factual knowledge in different depths, we split the concept of acquisition into three depths: (1) *memorization*: memorizing the exact sequence used for training (2) *semantic generalization*: generalizing the factual knowledge to a paraphrased format in a single-sentence level (3) *compositional generalization*: composing the factual knowledge presented in multiple sentences in the injected knowledge.

Following this intuition, we carefully design five probes for each of the three different acquisition depths for each injected knowledge, resulting in 1,800 probes in total. Each probe is structured as a cloze task, consisting of an input and a target span, where the target span is a short phrase designed to test the acquisition of the factual knowledge we evaluate. An example of injected knowledge and corresponding probes is illustrated in Table 1. All instances for the injected knowledge and probes are generated by prompting GPT-4 [2] using the definitions from the ECBD dataset [35] as a template, and filtering out invalid cases. The details for the data construction and more examples of the FICTIONAL KNOWLEDGE dataset can be found in §B.

**Evaluation metrics**    To conduct a detailed analysis of the LLMs' acquisition of factual knowledge during pretraining, we evaluate the model's state by examining log probabilities to obtain fine-grained information [41]. To quantitatively measure the trend of factual knowledge acquisition, we should first define the timestep where the local effect of updating the model using the injected knowledge completely pays off. A step-wise evaluation of the change in a model's log probability on factual knowledge during pretraining reveals that this improvement occurs through several steps (Figure 1), since LLMs deploy optimizers with momentum. Hence, we define the timestep where the log probability reaches a maximum value in a short interval after the model is trained on the injected knowledge, which we refer to as the *local acquisition maxima*.

**Definition 1** *Given a language model, let $\theta_t$ represent the model's parameters before the t-th update. Given injected knowledge k (used as a training instance) and the corresponding probe q (used as an evaluation instance), let $\ell(q; \theta)$ denote the log probability of the target span of q, provided by the model. Let a nonempty set $T_k = \{t_1, t_2, \ldots, t_n\}$ denote the steps where the model is updated with the minibatch containing the injected knowledge k, where $0 \leq t_1 < t_2 < \ldots < t_n$. Finally, let $t_w$ denote the window size. Then, the **local acquisition maxima** ($t_{LAM}(q, i)$) is defined as:*

$$t_{LAM}(q,i) = \operatorname*{argmax}_{t_i < t \leq t_i + t_w} \ell(q; \theta_t) \quad \text{where } t_i \in T_k. \tag{1}$$

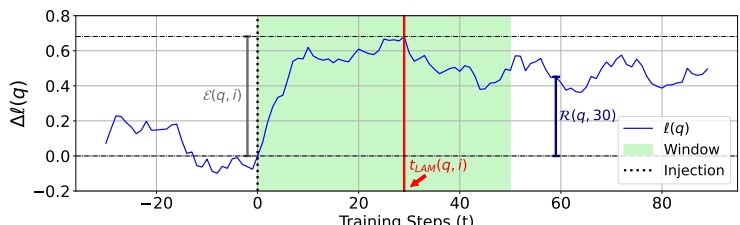

Figure 1: An illustration of the change of log probability of the target span of a probe ($\Delta\ell(q)$) measuring the memorization of factual knowledge on a short-term scale. At step 0 (marked as a dotted line), the model is trained with the injected knowledge which contains the factual knowledge evaluated by the probe $q$. The local acquisition maxima (marked as a red line) is the timestep where the log probability reaches its maximum within the window (shaded area), defined by $t_w$. The measurement of effectivity and retainability at $t = 30$ is visualized, where retainability is obtained by measuring the fraction of the purple line compared to the gray line.

In Eq.1, the definition of the local acquisition maxima is also dependent on the injected knowledge $k$ and the window size $t_w$, but we write $t_{\text{LAM}}(q, i)$ for brevity. We use the window size $t_w = 50$.[2][3]

Next, we define a metric to quantify the immediate improvement in the model's log probability of factual knowledge after it is presented with the knowledge for the $i$-th time. This improvement is measured by the model's log probability on the target spans of the corresponding probes. This metric, *effectivity*, will be used to answer the second research question.

**Definition 2** *Given a language model parameterized by $\theta$ trained with an injected knowledge $k$ at $t = t_i$ where $t_i \in T_k$, and a corresponding probe $q$, the **effectivity** ($\mathcal{E}(q, i)$) is defined as the absolute increase of the model's log probability on the target span of q between $t = t_i$ and $t = t_{LAM}(q, i)$, i.e.,*

$$\mathcal{E}(q, i) = \ell(q; \theta_{t_{LAM}(q,i)}) - \ell(q; \theta_{t_i}). \tag{2}$$

Finally, to investigate the forgetting phenomenon of acquired factual knowledge (RQ3), we define a metric that quantifies the fraction of improvement in log probability retained by the model after $t$ steps, relative to the local acquisition maxima of the last knowledge update.

**Definition 3** *Consider a language model parameterized by $\theta$ and trained with injected knowledge $k$ for $N$ iterations, occuring at timesteps $t_i \in T_k$ where $|T_k| = N$. Let $t_{pre}$ denote the last timestep before the model is first trained with $k$, i.e., $t_{pre} = \min(T_k)$. Given a corresponding probe $q$, **retainability** ($\mathcal{R}(q, t)$) is defined for $t \geq 0$ as follows:*

$$\mathcal{R}(q, t) = \frac{\ell(q; \theta_{t_{LAM}(q,N)+t}) - \ell(q; \theta_{t_{pre}})}{\ell(q; \theta_{t_{LAM}(q,N)}) - \ell(q; \theta_{t_{pre}})}. \tag{3}$$

Note that $\mathcal{R}(p, 0) = 1$ which represents that the factual knowledge is 100% retained at the local acquisition maxima of the last knowledge update. Additionally, $\mathcal{R}(p, t) = 0$ occurs when the log probability of the probe $p$ at $t_{SP(p)} + t$ equals that at $t_{pre}$. Thus, $\mathcal{R}(p, t) = 0$ indicates that the improvement in the log probability of factual knowledge, induced by updating the model with minibatches containing the injected knowledge at $t_{pre}$, is completely lost. This x-intercept of $\mathcal{R}(p, t)$ is crucial for interpreting the behaviors of LLMs, as will be discussed in detail in § 4.4. The measurement of the defined metrics are illustrated in Figure 1.

For the measurement of effectivity and retainability, we apply outlier detection using the IQR method with a factor of 1.5. This is particularly important for the measurement of retainability, as the small number of cases which showed no acquisition through training can give a very large value due to the very small denominator in Eq. 3.

**Knowledge injection during pretraining**    We explore how LLMs acquire and retain factual knowledge in terms of memorization and generalization by examining the following factors: (i)

---

[2]The $\beta_1$ of AdamW optimizer is configured to 0.9 in our experiments, implying that the contribution of the gradient of a given sequence to the momentum will be reduced to approximately $0.9^{50} \approx 0.0052$ after 50 steps. Therefore, $t_w = 50$ is a reasonable choice for the window size.

[3]If optimizers without momentum (*e.g.*, RMSProp) are used, the local effect of training the model at timestep $t$ will be fully reflected immediately after that step. In such cases, $t_w$ should be 1 and $t_{LAM}$ will reduce to $t + 1$.

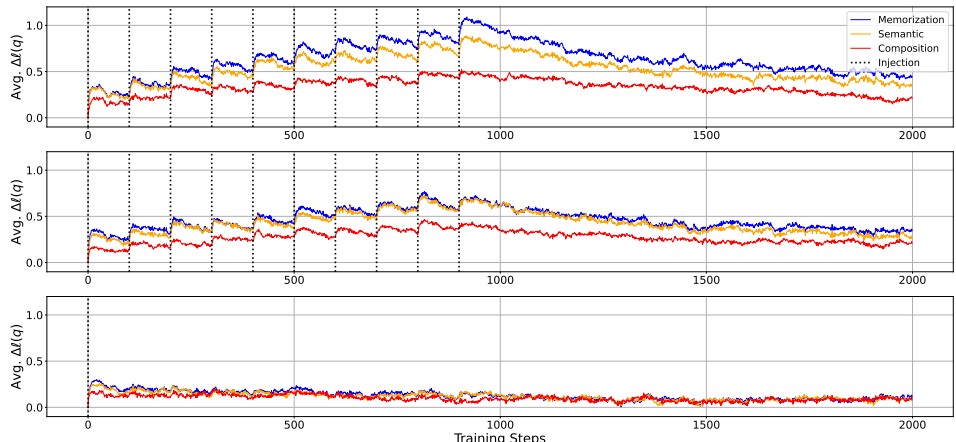

Figure 2: Change in the average log probability of target spans of the probes plotted against training steps during the continuation of pretraining OLMo-7B *mid* checkpoint (trained on 500B tokens) with injecting the knowledge in the FICTIONAL KNOWLEDGE dataset. Results are shown for *duplicate* (**Top**), *paraphrase* (**Center**), and *once* (**Bottom**) injection scenarios. Note the immediate and distinctive increase of log probability after the model is updated with the injected knowledge, marked by dotted vertical lines.

varying knowledge injection scenarios (*duplication*, *paraphrase*, *once*), (ii) varying pretraining stages (*early*, *mid*, and *late*, pretrained with approximately 170B, 500B, and 1.5T tokens, respectively), (iii) varying model sizes (1B and 7B), and (iv) varying training batch sizes (2048 and 128). To this end, we resume pretraining OLMo [21] intermediate checkpoints restoring the optimizer and scheduler states the same way OLMo is pretrained, using the pretraining data of OLMo (Dolma v1.5 [43]), except that we inject factual knowledge every 100 training steps by replacing a part of original pretraining batch with the injected knowledge of the FICTIONAL KNOWLEDGE dataset.[4] Each injected knowledge is short enough to fit into one pretraining sequence in the batch, and we fill the rest of the sequence with the original sequence in the batch. To investigate the difference in the factual knowledge acquisition dynamics when the models are presented with the knowledge, we inject factual knowledge with three different injection scenarios: *duplication*, *paraphrase*, and *once*. For the *duplication* injection scenario, we inject the same knowledge 10 times with an interval of 100 training steps. In the *paraphrase* injection scenario, we inject paraphrased knowledge instead of showing identical sequences, every time it is presented to the model. Lastly, in the *once* injection scenario, we inject the knowledge only once at the start of the training. After the injection is complete, we continue pretraining as normal. The details for the training setup can be found in §D.

## 4 Results

### 4.1 Factual knowledge acquisition occurs by accumulating the observations of the fact

Figure 2 shows the progress of factual knowledge acquisition of OLMo-7B, by averaging the model's log probability across the target spans of the probes for each injection scenario, evaluated at each training step. Regardless of the acquisition depths (memorization, semantic generalization, and compositional generalization), the model's log probability measured on the probes shows an immediate and distinctive increase, after the model is updated with the batch containing the injected knowledge. However, the log probability decreases again, as the knowledge is not presented to the model afterward. This observation directly demonstrates the mechanism of factual knowledge acquisition: **LLMs acquire factual knowledge by accumulating micro-acquisitions with subsequent forgetting each time the model encounters the knowledge during pretraining.**

Several findings can be further obtained from Figure 2. First, when the model is updated after seeing the factual knowledge, the most significant improvement in log probability is observed for memorization, followed by semantic generalization, and the least improvement is seen in compositional generalization. Next, however, the gap between memorization and semantic generalization almost

---

[4]We use OLMo for the experiments since the intermediate checkpoints, optimizer states, and batch sequence data for pretraining the model are made publicly available.

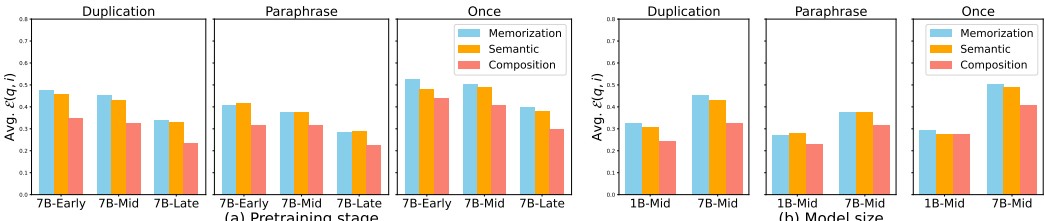

Figure 3: Effectivity averaged across various probes and each time of injection, measured for different injection scenarios, and acquisition depths. Note that the effectivity does not improve as the model is trained with more tokens (**Left**), whereas there is a clear improvement as the model size scales (**Right**).

disappears in the *paraphrase* injection scenario. Third, when the model is updated with the *duplication* injection scenario, the model shows a larger improvement of log probability in all acquisition depths, but also the forgetting is faster, eventually resulting in a similar level of improvement at the end of the training ($t = 2000$) compared to the *paraphrase* injection scenario.

These patterns are consistent across all pretraining stages of OLMo-7B we investigate (§E.1). Intriguingly, the training dynamics of OLMo-1B *early* checkpoint (Appendix Figure 8) show much more unstable dynamics than those of later checkpoints (Appendix Figure 9 and 10) and the *early* checkpoint of OLMo-7B (Appendix Figure 6). The distinctive behavior of the OLMo-1B *early* checkpoint suggests that pretraining on a certain number of tokens may be required for the model to acquire factual knowledge stably and that such a threshold may be higher for smaller models.

## 4.2   Effects of model scale and pretraining stage on knowledge acquisition dynamics

Next, we measure effectivity (Eq. 2) to quantify the improvement of the LLMs' log probability after being trained with the injected knowledge, averaged across all probes ($q$) and encounters ($i$). The results are demonstrated in Figure 3. The average effectivity is the largest in the *Once* injection scenario since the effectivity is higher when the model encounters the injected knowledge for the first time, which is further discussed in §H.

In all injection scenarios, there is an improvement in effectivity when the model size is scaled from 1B to 7B (as shown on the right side of Figure 3).[5] On the other hand, surprisingly, the effectivity of fact acquisition does not improve with checkpoints trained with more tokens, as shown on the left side of Figure 3. This tendency is consistent across all model scales and injection scenarios (see also Appendix Figure 11). Moreover, this tendency is not attributed to training the models with a decreased learning rate through learning rate decay, as demonstrated by an additional experiment of training three checkpoints using the same constant learning rate. The results with the constant learning rate show that effectivity does not significantly improve in the checkpoints of later stages of pretraining where more pretraining tokens are seen (§F). Therefore, the observation implies that the effectivity of LLMs in acquiring factual knowledge does not significantly improve throughout the progress of pretraining.

While our finding that effectivity remains unchanged for different stages of pretraining may seem contradictory to the widely known observation that the amount of pretraining data is a critical factor in the performance of LLMs [23, 27], we suggest a plausible hypothesis based on further observations in §4.3. Specifically, we suggest that the high performance of LLMs trained with larger and more diverse datasets is not primarily due to an emergent ability from the sheer amount of tokens observed during training [50], but rather because the model encounters a wider variety of knowledge more times, which allows for the accumulation of log probabilities of more knowledge become high enough to be decoded as outputs of the model. We discuss this hypothesis further in §4.4.

Comparing the *duplication* and *paraphrase* injection scenarios, the *duplication* injection scenario naturally shows higher effectivity for memorization. However, the higher effectivity in the *duplication* injection scenario for semantic generalization and compositional generalization appears to be

---

[5]For a fair comparison of the effectivity of the 1B and 7B models, the OLMo-1B *Mid* checkpoint is trained using the same initial learning rate as the OLMo-7B *Mid* checkpoint (the specific value is provided in Appendix Table 5). The measured effectivity for all OLMo-1B checkpoints with the original learning rate is presented in Appendix Figure 11.

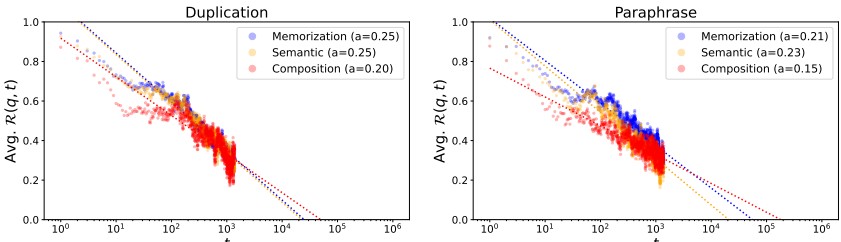

Figure 4: Average retainability against training steps past the local acquisition maxima, measured with OLMo-7B *mid* checkpoint. The x-axes are in log scale. **Left**: *duplication*. **Right**: *paraphrase*.

Table 2: Decay constant of average retainability ($\mathcal{R}(p, t)$) measured with OLMo-7B at different pretraining stages, acquisition depths, and injection scenarios. Note that the larger value indicates that the model forgets acquired knowledge with a higher rate.

| Pretraining stage | | Early (170B) | Mid (500B) | Late (1.5T) |
|---|---|---|---|---|
| **Duplication** | Memorization | $0.26_{\pm 0.0020}$ | $0.25_{\pm 0.0019}$ | $0.20_{\pm 0.0019}$ |
| | Semantic | $0.24_{\pm 0.0018}$ | $0.25_{\pm 0.0022}$ | $0.21_{\pm 0.0021}$ |
| | Composition | $0.18_{\pm 0.0020}$ | $0.20_{\pm 0.0032}$ | $0.16_{\pm 0.0024}$ |
| **Paraphrase** | Memorization | $0.20_{\pm 0.0019}$ | $0.21_{\pm 0.0023}$ | $0.18_{\pm 0.0022}$ |
| | Semantic | $0.20_{\pm 0.0020}$ | $0.23_{\pm 0.0024}$ | $0.21_{\pm 0.0024}$ |
| | Composition | $0.14_{\pm 0.0025}$ | $0.15_{\pm 0.0022}$ | $0.19_{\pm 0.0030}$ |

counterintuitive, as it is widely observed that deduplication of pretraining data is an important factor in improving model performance [29, 52]. In the following sections, we will address this question by demonstrating that the models exhibit faster forgetting in generalizing factual knowledge when presented with duplicated texts (§4.3).

## 4.3 Forgetting in factual knowledge acquisition

**Training steps and the forgetting of acquired factual knowledge have a power-law relationship**
The exponential trend of forgetting has been reported in various aspects of LLM training, including memorization in pretraining [46] and task performances in continual learning [33, 39]. Motivated by this, we investigate whether the exponential trend of forgetting persists in the context of factual knowledge acquisition in LLM pretraining. Figure 4 illustrates the trend of retainability against the training steps past the local acquisition maxima. We find that the trend of $\mathcal{R}(p, t)$ against $log(t)$ fits a linear function very well ($R^2 > 0.80$ for memorization and semantic generalization, and $R^2 > 0.65$ for compositional generalization). This trend is persistent across all acquisition depths, and all training conditions (§E.4 and §E.5). Guided by empirical observations, we model the trend of forgetting using a power-law model in further investigations.

**How quickly is the acquired factual knowledge lost?** The absolute value of the slope of the fitted lines in Figure 4 can be interpreted as the decay constant ($a$) of retainability, formally,

$$\Delta \mathcal{R}(p, t) \approx -a \cdot \log\left(\frac{t_2}{t_1}\right) \quad \text{for } 0 < t_1 < t_2 < \tau, \quad \text{where } \mathcal{R}(p, \tau) = 0 \text{ and } a > 0. \quad (4)$$

Thus, the measured decay constant represents how fast (in terms of fraction) the model loses the improvement of log probability. Table 2 shows the decay constants of retainability measured for three OLMo-7B intermediate checkpoints, for *duplication* and *paraphrase* injection scenarios.

There are several observations in Table 2. First, the forgetting in compositional generalization is slower (the decay constant $a$ is smaller) than in memorization and semantic generalization. Combined with the observations in previous sections, the acquisition of compositional generalization accumulates most slowly but is more robust to forgetting. Second, the forgetting tends to be slower in the *paraphrase* injection scenario compared to the *duplication* injection scenario. This finding will be further discussed in §4.4, regarding the importance of deduplicating training data. Finally, the decay constants are similar for the two earlier checkpoints but smaller for the *late* checkpoint in the *duplication* injection scenario. We demonstrate that this is due to the reduced learning rate from

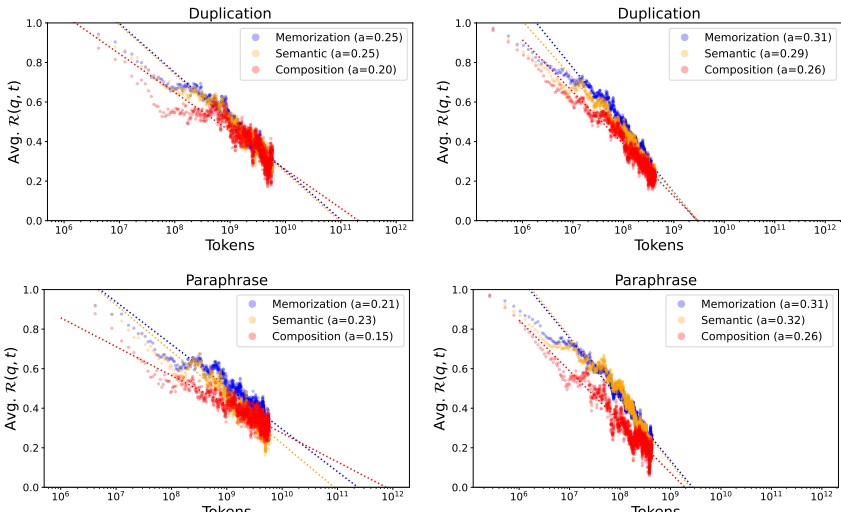

Figure 5: Comparison of the forgetting dynamics of pretraining (**Left**) and training with reduced batch size (**Right**), measured with OLMo-7B *mid* checkpoint. Note that the x-axis represents the number of training tokens instead of training steps, which has a shifting effect on the data plotted in Figure 4.

learning rate scheduling (Appendix Table 5), as the decay constants show no decrease for the later checkpoint when each checkpoint is trained with the same constant learning rate (Appendix Table 9).

**Pretraining with a larger batch size helps LLMs acquire more knowledge**   It is a common practice to pretrain LLMs with a very large batch size to leverage parallel computing [13, 21, 25, 30, 49]. However, the effects of increasing training batch size in terms of the LLMs' acquisition of factual knowledge remain underexplored. In this section, we examine whether pretraining LLMs with a larger batch size is advantageous regarding factual knowledge acquisition. Specifically, we continue training LLMs with a batch size reduced by a factor of 16 compared to the original pretraining batch size, *i.e.*, from 2048 to 128.

Figure 5 compares the forgetting dynamics of OLMo-7B *mid* checkpoint between pretraining and training with the reduced batch size. The results have several implications for the advantage of pretraining LLMs with a larger batch size. First, comparing Figure 3 and Appendix Figure 21, LLMs trained with the smaller batch size show higher effectivity. However, the decay constant tends to be higher, comparing the numbers in Table 2 and Appendix Table 10. Furthermore, the anticipated x-intercept is significantly decreased by dozens of times, comparing Appendix Table 6 and 11. This implies that the models trained with smaller batch sizes have shorter *learnability threshold*, the point such that an LLM cannot learn the knowledge presented with intervals longer than that threshold, which we discuss in detail in the following section (§4.4). In other words, when an LLM is trained with a smaller batch size, factual knowledge should be presented more often to the model so as not to be forgotten and the set of *learnable* knowledge is reduced. Second, accelerated forgetting with a smaller batch size is more pronounced for compositional generalization compared to memorization and semantic generalization. In brief, the results suggest that pretraining with a small batch size reduces the set of learnable knowledge due to accelerated forgetting, and leads to worse compositional generalization performance of learned factual knowledge.

## 4.4 Implications for LLM pretraining

**Why is popularity important for factual knowledge acquisition?**   The estimated x-intercepts in Figure 5 represent the number of additional training tokens that would lead to the complete loss of the factual knowledge acquired by training.[6] Hence, if a given factual knowledge in the pretraining dataset is in the long-tail and the knowledge is presented to the model with an interval longer than a certain threshold, such knowledge will be impossible to be decoded as the top-k generation of

---

[6]The exact values of the estimated x-intercepts can be found in Appendix Table 6.

the model, or *learned*, regardless of the duration of the pretraining.[7] This implies that there is a *learnability threshold*, a threshold of the interval where the model fails to acquire knowledge of which its encounter interval is longer than the threshold. Most well-known facts are likely to be presented to the model with an interval of the training steps shorter than this *learnability threshold*. In such a case, the model will accumulate the increased log probability of the knowledge upon each encounter of the knowledge as the pretraining progresses, and at some point, the accumulated log probability of the knowledge will be high enough to generate the knowledge as the decoding output of the model [41]. Moreover, LLMs will accumulate the log probability faster for more popular knowledge, and thus the acquisition of such knowledge will be reflected in the model's top-k output sequence generation in a relatively earlier pretraining stage, as demonstrated in [8].

In summary, we hypothesize that the popularity of the knowledge in the pretraining data influences how quickly this knowledge begins to be 'revealed' in the generated sequences during pretraining, except for the knowledge in the long-tail whose low popularity makes the encounter interval longer than the *learnability threshold*. Also, as briefly mentioned in §4.2, we hypothesize that the reason why larger and more diverse pretraining data helps the model performance is that the model can acquire a broader range of factual knowledge (more knowledge will be presented with an interval shorter than the *learnability threshold*) since the skewness of the distribution of factual knowledge popularity is likely to be mitigated as the data becomes larger and more diverse.

**Why does deduplication enhance model performance?**    Recent pretraining corpora are thoroughly deduplicated [9, 28, 38, 43, 47, 48], as it is widely observed that data deduplication can improve model performance [1, 29, 42, 52]. Our results suggest that the smaller decay constant in the *paraphrase* injection scenario observed in §4.3 can explain the advantages of training LLMs with deduplicated training data, as deduplication tends to slow the forgetting of generalizing acquired factual knowledge. This can also be observed in Figure 2, as the gap of the increase of log probability immediately after encountering the injected knowledge is large between the *duplication* and *paraphrase* injection scenarios, but this gap diminishes at the end of the measurement. Moreover, since the model tends to provide a higher increased log probability to the memorization rather than generalization (Figure 2 and 3), presenting the model with duplicated texts with a short interval will result in the widening of the gap between memorization and generalization, which will drive the model to prefer generating memorized contexts compared to generalizing factual knowledge [4].

## 5    Discussion and Conclusions

In this work, we study how LLMs acquire factual knowledge during pretraining. Our findings and contributions can be summarized as follows:

- We propose methods, datasets, and metrics for performing a fine-grained analysis of factual knowledge acquisition dynamics during LLM pretraining.

- We demonstrate that factual knowledge acquisition in LLM pretraining is achieved through accumulating micro-acquisitions, each of which occurs whenever the model is updated after seeing the factual knowledge. When the model is not presented with factual knowledge, forgetting occurs and the acquisition of the knowledge is gradually diluted.

- However, while the amount of immediate improvement in log probability upon observation of the knowledge increases for larger models, the amount does not significantly increase throughout the progress of pretraining. This finding suggests that the benefits of scaling the model size and pretraining tokens are qualitatively different.

- There is a power-law relationship between training steps and forgetting of acquired factual knowledge, in terms of both memorization and generalization. Also, pretraining LLMs with deduplicated data and larger batch sizes enhances the acquisition of factual knowledge, making them more robust against forgetting the learned factual knowledge.

---

[7]This *theoretical* threshold may not be equal to the estimated x-intercepts presented in Figure 5, as we estimate the threshold based on the controlled experiment of injecting factual knowledge. In addition, the actual learnability threshold is likely to vary for different types of factual knowledge due to several factors, such as the number of similar/related facts or temporal conflicts in the pretraining data.

- We provide potential explanations for recently observed, yet underexplored behaviors of LLMs. First, we propose that the improved performance of LLMs through data scaling results from consistent improvements rather than an emergent ability to acquire factual knowledge more quickly during pretraining. Second, we hypothesize that LLMs struggle to acquire unpopular knowledge because they need sufficient exposure to factual knowledge with intervals shorter than the *learnability threshold* to increase the probability. Third, our findings suggest that deduplicating the pretraining corpus improves LLM performance by preventing the model from assigning a higher probability to duplicated sequences and helping it retain acquired generalization longer.

Overall, we demonstrate the importance of understanding the factual knowledge acquisition dynamics of LLMs to understand the behavior of LLMs, opening up a promising avenue for future research.

## Acknowledgments

We would like to thank Seongyun Lee, Suehyun Park, Hyeonbin Hwang, Geewook Kim, Juyoung Suk, Aengus Lynch, and Katja Filippova for their valuable feedback on our work.

This work was supported by Institute for Information & communications Technology Promotion(IITP) grant funded by the Korea government(MSIT) (No.RS-2019-II190075 Artificial Intelligence Graduate School Program(KAIST).

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

# Appendix

## A    Limitations

Although they do not affect the findings and implications of our work, there are several limitations. First, we do not perform evaluations based on the generation output of the model, and we do not investigate the exact relationship between the model's accumulation of probability of factual knowledge and the model's generation output. Second, we do not analyze the pretraining dynamics at very early stages, which can exhibit significantly different behaviors [24]. Third, we do not study the effect of training batch size and learning rate on the dynamics of factual knowledge acquisition across multiple values. Future works exploring these would help us to further enhance our understanding of LLMs.

## B    Dataset Construction and Examples

We construct a FICTIONAL KNOWLEDGE dataset by prompting GPT-4 [2] with C.1 to generate descriptions for non-existent, fictional entities using the format of the ECBD [35] dataset, which is based on English Wikipedia articles. We select only the generated descriptions of the fictional entities that can produce at least five sentences suitable for a cloze task when the last span of the sentence is set as the target label. We repeat this until a total of 120 descriptions are produced. We call this "injected knowledge" in this paper. This process facilitates us to investigate the factual knowledge acquisition of the language models in a more controlled setup, as we can ensure that the model has never encountered the facts contained in the injected knowledge during the pretraining process. For the *paraphrase* injection training scenario mentioned in §3, we generate 9 paraphrased injected knowledge for each original injected knowledge by prompting GPT-4 with C.2.

The types of probes for the injected knowledge consist of *memorization* probes, *semantic generalization* probes, and *compositional generalization* probes. For each injected knowledge, 15 probes are generated, with 5 for each type. First, the *memorization* probes are constructed by extracting exact sentences from the injected knowledge that ends with a named entity and setting the named entity as the target span. Next, the *semantic generalization* probes are created by prompting GPT-4 with C.3 to paraphrase each memorization probe while maintaining the target span and requiring no additional context. Lastly, *compositional generalization* probes are created by prompting GPT-4 with C.4 to create cloze tasks to evaluate whether new factual knowledge can be inferred by integrating and generalizing the factual knowledge in the injected knowledge. We constrain that the *compositional generalization* probes should avoid lexical overlap with the injected knowledge as much as possible and should not require additional context beyond the knowledge in the injected knowledge. To ensure the validity of the generated *compositional generalization* probe sets, we ask GPT-4 using prompt C.5 to evaluate whether each probe meets these conditions, answering with "yes" or "no". Only the probes that receive a "yes" response are selected. Examples of injected knowledge and paraphrased injected knowledge from the FICTIONAL KNOWLEDGE dataset can be found in Table 3 and the *memorization* probes, *semantic generalization* probes, and *compositional generalization* probes used to evaluate the acquisition of knowledge can be found in Table 4.

Table 3: An example of injected knowledge and paraphrased injected knowledge in the FICTIONAL KNOWLEDGE dataset.

| | |
|---|---|
| **Injected Knowledge** | The fortieth government of Mars, or the ZorgonŽ2013Calidus government, was officially constituted on 5 Outcrop 6678, following the interplanetary governance elections held that Martian cycle. Zorgon, a renowned Martian statesman, was a prominent figure that took office as Prime Minister, being a central character in Martanian politics before the formation of this government. Calidus, on the other hand, served as the governmental second-in-command, known for his in-depth knowledge of astropolitics, which enhanced the efficiency of the ZorgonŽ2013Calidus government. Mars, historically known for its centralised sub-planet distribution, underwent significant political reform under Zorgon's leadership. The ZorgonŽ2013Calidus government, on August cycling in the same Mars year, introduced more devolved power structures across its 50 provinces, an unprecedented move in Martian politics. A parallel development was the Calidus-led diplomatic initiative with the moon, Deimos. The initiative was a landmark effort to establish interplanetary ties, an essential aspect of the ZorgonŽ2013Calidus agenda. The democratic system of Mars, which was entering a transitory phase, picked up speed under the ZorgonŽ2013Calidus government. Mars, on 13 Amazonis 6678, saw a radical shift in its governance with the introduction of new legislative paradigms. The ZorgonŽ2013Calidus government on 22 Amazonis, successfully passed five bills that redefined Martanian healthcare, reflecting the administration's commitment. The ZorgonŽ2013Calidus government also prioritised interplanetary relations. Mars, by the end of 6679 Martian cycle, had set up embassies on Jupiter's moons Ganymede and Callisto. Zorgon's far-reaching vision was mirrored in these concrete steps to ensure the wellbeing of Mars' inhabitants. The MarsŽ2013Ganymede Pact, a resultant product of this diplomatic effort, was officially signed on 12 Tharsis 6680. Zorgon, in alignment with increasing demands for technological advancements, launched the interplanetary digital initiative on 7 Olympus 6680. Mars, under the ZorgonŽ2013Calidus government, showed tremendous growth in the field of Martian technology. Calidus, in his final public address on 31 Tharsis 6682, emphasised the administration's focus on sustainable development, reflecting a commitment to Martian environmental health. The ZorgonŽ2013Calidus government, despite facing several political challenges, remained resilient till the end of its term. The Fortieth Martian Council was effectively disbanded on 19 Hellas 6684. The ZorgonŽ2013Calidus government's tenure is remembered as a landmark period in the history of Martian governance. |
| **Paraphrased Injected Knowledge** | The Zorgon-Calidus administration, also known as the 40th Martian government, was established after the interplanetary elections on 5 Outcrop 6678. Zorgon, notable for his esteemed political career, assumed office as the Prime Minister while Calidus, distinguished for his understanding of astropolitics, acted as his deputy. This tag-team transformed Mars' traditional centralized governance by decentralizing power across its 50 provinces in August of the same Martian year. Concurrently, Calidus spearheaded a diplomatic initiative with one of Mars' moon's, Deimos, strengthening interplanetary relations. The newly refurbished democratic system gained momentum under the leadership of the Zorgon-Calidus administration. A milestone in this transition was marked on 13 Amazonis 6678 when Mars adopted new legislative standards. On 22 Amazonis, the government also passed five bills improving Martanian healthcare. Interplanetary diplomacy remained high on the agenda, with Mars establishing embassies on Ganymede and Callisto, Jupiter's moons, by the end of 6679. The interplanetary agreement, known as the Mars-Ganymede Pact, was formally signed on 12 Tharsis 6680. Aligning with the demand for progressive technology, Zorgon inaugurated the interplanetary digital initiative on 7 Olympus 6680 causing significant technological development on Mars. In his last address to the public on 31 Tharsis 6682, Calidus stressed the significance of sustainable growth on Mars. The Zorgon-Calidus administration despite opposition, fulfilled its term resolutely until its disbandment as the 40th Martian Council on 19 Hellas 6684. The Zorgon-Calidus era is regarded as a pivotal period in Martian history. |

Table 4: An example of probe sets in the FICTIONAL KNOWLEDGE dataset. The **target span** of each probe is bolded.

| | |
|---|---|
| **Memorization probes** | The fortieth government of Mars, or the ZorgonŽ2013Calidus government, was officially constituted on 5 Outcrop 6678, following the interplanetary governance elections held that **Martian cycle** |
| | Mars, historically known for its centralised sub-planet distribution, underwent significant political reform under **Zorgon's leadership** |
| | The democratic system of Mars, which was entering a transitory phase, picked up speed under the **ZorgonŽ2013Calidus government** |
| | Mars, by the end of 6679 Martian cycle, had set up embassies on Jupiter's moons Ganymede and **Callisto** |
| | Zorgon's far-reaching vision was mirrored in these concrete steps to ensure the wellbeing of **Mars' inhabitants** |
| **Semantic probes** | The ZorgonŽ2013Calidus government, also known as the fortieth government of Mars, was formally established on 5 Outcrop 6678, after the elections for interplanetary governance took place during that **Martian cycle** |
| | Mars, previously recognized for its focused distribution of sub-planets, experienced substantial political transformation during **Zorgon's leadership** |
| | The progression towards a transitory phase accelerated in the democratic system of Mars under the rule of the **ZorgonŽ2013Calidus government** |
| | By the conclusion of the 6679th Martian cycle, Mars had established diplomatic embassies on two of Jupiter's moons, Ganymede and **Callisto** |
| | The expansive outlook of Zorgon was reflected in these tangible measures taken to safeguard the welfare of **Mars' inhabitants** |
| **Composition probes** | The diplomatic initiative to establish interplanetary ties had a historic agreement with one of Mars' moons, namely **Deimos** |
| | ZorgonŽ2013Calidus government rapidly expedited the transitory phase of the Martian **democratic system** |
| | Besides domestic policies, the Zorgon-Calidus government was known for fostering abroad relationships which was evident from their establishment of embassies on Jupiter's moons, namely **Ganymede and Callisto** |
| | The repercussion of their diplomacy with the moons of Jupiter was reflected in a formal agreement termed **the MarsŽ2013Ganymede Pact** |
| | Keeping up with the global emphasis on technology, the ZorgonŽ2013Calidus government launched the **interplanetary digital initiative** |

## C Prompts Used for Dataset Generation

### C.1 Prompts for the generation of injected knowledge

Carefully read the provided sentence; this is a short passage
containing factual knowledge, that is extracted from Wikipedia:\n\n
{DEFINITION IN ECBD DATASET}\n\nNow, assume that you are writing a very
long and detailed descriptive paragraphs (more than 20 sentences) using
the provided passage as a template. However, you should replace the
named entities(person, country, act, etc.) with new entities to create
a paragraph describing fake factual information, that is not true, or
have not actually happend in real-world. Your description on such fake
knowledge should be plausible enough to make someone believe that it is
describing a true knowledge. You should always start and finish every
sentence with a named entity. Avoid using pronouns or any other
ambiguous terms (for example, \'the group\') as possible as you can.
Finally, avoid to generate knowledge that is potentially harmful. Avoid
generating fake knowledge that contains prejudices, discrimination
on any kind of social groups. Output the created paragraph only.\n\n

### C.2 Prompts for the generation of paraphrased injected knowledge

The following text needs to be paraphrased to convey the same meaning
in different words:\n\n\"{ORIGINAL INJECTED KNOWLEDGE}\"\n\nPlease
paraphrase the above text clearly and concisely.

### C.3 Prompts for the generation of semantic generalization probes

Paraphrase the provided text with a constraint: the paraphrased
sentence should be ended with the specified target, where the original
sentence also ends with the target. Note that the paraphrased sentence
should be semantically equivalent to the original sentence, and it
should not contain any additional factual knowledge, nor lacks any
factual knowledge that is stated in the original text. In addition, the
content of the paraphrased text should be able to be fully understood
without any ambiguity.\n Here are some exmaples:\n\n[Example1 1]\n\n
Input: The Lionheart Battalion (LB) is a fictitious white nationalist
militia group in Spain.\nTarget: Spain\nOutput: The Lionheart Battalion
(LB) is a fictional militia group with white nationalist beliefs
located in Spain.\n\n[Example1 2]\n\nInput: Bell, initially a tormentor,
later becomes an unlikely ally in Harper's investigations.\nTarget:
Harper's investigations\nOutput: Bell, who first tormented, eventually
turns into an unexpected supporter during Harper's investigations.
\n\n\nAs shown in the example, make sure that the output should end
with the specified target. Never finish the sentence with any other
words.\n\nNow, this is your input and target:\n\nInput:
{MEMORIZATION PROBE}\nTarget: {TARGET FOR MEMORIZATION PROBE}\nOutput:

### C.4 Prompts for the generation of compositional generalization probes

You are tasked with evaluating a participant's intelligence(in terms of
generalization, composition, and inference) by measuring their ability
to understand and combine the implications of different factual
knowledge presented in a passage and apply them to deduce unseen
knowledge. Specifically, you will create a next-word prediction task
consisting of inputs and targets. The objective is to assess whether
the participant can integrate and generalize the implications of the
factual knowledge from the passage, combining different pieces of
information to infer new factual knowledge.\n\nThe target should

consist of less then five words that complete the sentence when combined with the input, where the input is an incomplete sentence. The inputs and targets must be designed so that the target can only be accurately answered if the participant can perform complex generalization and integration based on the provided knowledge.\n\n Create eight different pairs of inputs and corresponding targets that require the participant to combine various factual knowledge presented in the passage, to deduce unseen knowledge. Avoid lexical overlaps with the passage as much as possible. Also, the content in the task should not ask for factual knowledge that is directly mentioned in the given passage, in other words, difficult enough. Additionally, ensure that the input and target can be understood and answered without additional context, assuming that the reader has comprehended and remembered the knowledge from the passage. Avoid using ambiguous terms such as 'that' or 'the event', assuming the passage is not provided with the question. Finally, most importantly, be creative as much as you can.\n\nPlease present your answers in the following format:\n\nProbe1: [YOUR_PROBE_ENDS_WITH_AN_UNDERSCORE]\nAnswer1: [YOUR_ANSWER_TO_THE_PROBE]\n\nNow, this is your passage:\n\n {ORIGINAL INJECTED KNOWLEDGE}

## C.5 Prompts for the validation of generated compositional generalization probes

You will be provided with a pair of cloze-task question and answer, and the problem's goal is to evaluate the subject's factual knowledge. Your task is to verify whether the provided pair of question and answer is properly designed to evaluate the factual knowledge. Assume that the subject has been already informed with the counterfactual knowledge before. Then, we are testing the subject's counterfactual knowledge. Note that regardless of the consistency of the factual knowledge tested in the problem, we say that the problem is properly designed if there is no ambiguity in the question and answer. So the question is verifying: Can the content of the question be fully understood and properly answered without any ambiguity or the need of additional context, given that the corresponding factual knowledge is existent?\n \nAfter providing your explanation, you should give your answer in 'yes' or 'no'. The answer should be 'yes' only if both of the conditions are satisfied, and the answer should be 'no' otherwise.\n For example, this is an example of your answer:\n\nExplanation: [YOUR_EXPLANATION]\nAnswer: [YES_OR_NO]\n\nHere are some example inputs and answers:\n\n[Example 1]\nQuestion: Within the realm of fantasy, he is ranked second in command in the _____\nAnswer: Lionheart Battalion\n \nExplanation: The example provided is not entirely clear or straightforward in its design to evaluate factual knowledge. The question, \"Within the realm of fantasy, he is ranked second in command in the _____,\" contains a few ambiguities. Firstly, \"the realm of fantasy\" is a broad and non-specific term, which could refer to any number of fantasy stories, games, or universes. Secondly, the phrase \"he is ranked second in command\" does not specify who \"he\" refers to, nor does it establish a clear context or a specific entity to which the answer \"Lionheart Battalion\" could logically be connected without additional information. This lack of specificity and context does not allow the question to be answered accurately based solely on factual knowledge without guessing or assuming additional context. The problem does not provide enough information to identify which fantasy setting is being referred to, nor does it give any clues about the character or the organizational structure within which this character operates.\n Answer: no\n\n[Example 2]\nQuestion: Jaccard Hume was the first person to land on _____\nAnswer: Mars\n\nExplanation: This question and answer

pair seems straightforward and specific in its design to evaluate factual knowledge. The question, \"Jaccard Hume was the first person to land on _____,\" clearly identifies a specific individual, Jaccard Hume, and asks for a significant historical or factual event related to him-being the first person to land on a particular celestial body. The answer provided is \"Mars,\" which is clear and direct. Assuming the subject has the necessary factual knowledge about Jaccard Hume and his achievements, there is no ambiguity in either the question or the answer. The answer \"Mars\" directly fills the blank without the need for additional context or interpretation. Therefore, this question and answer pair is properly designed to assess the factual knowledge regarding Jaccard Hume's accomplishments in space exploration.\nAnswer: no\n\nNow, here is the input text:\n\nQuestion: {GENERATED COMPOSITIONAL GENERALIZATION PROBE} _____Answer: {GENERATED TARGET OF COMPOSITIONAL GENERALIZATION PROBE}\n\n

# D   Detailed Training Setup

To continue training almost similar to the pretraining setup, we use OLMo[21], as it provides not only intermediate model checkpoints but also the exact sequence of data instances used for pretraining, the optimizer state, and the learning rate scheduler. Throughout the entire pretraining process, the language model is trained with a language modeling objective.

Except for the batches that include injected knowledge from FICTIONAL KNOWLEDGE dataset at specific step intervals, we train OLMo with batches from the Dolma corpus [43] in the same order which is used in OLMo pretraining. Specifically, we load the training batch that OLMo will be seen at the specific pretraining step, append the injected knowledge from the FICTIONAL KNOWLEDGE dataset to the front of each row, and truncate the original rows from the end by the token length of the injected knowledge. This approach creates batches that have the same size as the original pretraining batches, with 2048 rows and a sequence length of 2048, meaning each batch contains 4M tokens. We adopt this method to deviate as little as possible from the original pretraining data distribution.

In the FICTIONAL KNOWLEDGE dataset, which consists of 120 descriptions of fictional knowledge, we use the first 1-40 injected knowledge to examine the dynamics of knowledge acquisition in the *paraphrase* injection scenario which is described in §3. The 41-80 injected knowledge is used for the *duplication* injection scenario, and the 81-120 injected knowledge is used for the *once* injection scenario.

For each injection scenario, the FICTIONAL KNOWLEDGE data are injected into the batch and trained according to the following rules. In the *duplication* injection scenario, injected knowledge in the FICTIONAL KNOWLEDGE dataset is injected into the original pretraining batch, and the language model is trained on this modified batch 10 times every 100 steps. Next, in *paraphrase* injection scenario, similar to the *duplication* injection scenario, the model is trained on the modified batches containing FICTIONAL KNOWLEDGE every 100 steps for a total of 10 times, however, in this case, paraphrased injected knowledge is used at each injection step. Lastly, in the *once* injection scenario, the modified batch containing injected knowledge of FICTIONAL KNOWLEDGE is shown to the language model just once, after which it continues training on the original batch of Dolma corpus.

After 1000 steps of pretraining following the above rules, an additional 1500 steps of pretraining are conducted using the Dolma corpus for experiments analyzing forgetting dynamics in §4.3. The Dolma corpus used at these steps is a corpus that will be viewed starting from the 360,000th step of pretraining the OLMo. This approach ensures consistency in the Dolma corpus across all conditions while guaranteeing that the corpus has not been seen in any previous pretraining processes. Continued pretraining of a total of 2500 steps takes approximately 3 days using 8 80GB A100 GPUs.

To examine the differences in knowledge acquisition dynamics based on model size, we use OLMo-7B and OLMo-1B. For differences based on the number of pretrained tokens, we use intermediate checkpoints at **Early (170B)** stage (specifically, 177B tokens for 7B and 168B tokens for 1B), **Mid (500B)** stage (specifically, 500B tokens for 7B and 494B tokens for 1B), and **Late (1.5T)** stage (1.5T tokens for 7B and 1B). Since the initial checkpoints of OLMo-1B are stored in units of 10000, it is the best choice in the given situation to select the checkpoint trained with the number of tokens closest to 177B. The differences in initial learning rate values for each case based on different model sizes and pretraining stages are recorded in Table 5 below.

Table 5: The initial learning rate for each intermediate OLMo checkpoint based on model sizes and the pretraining stages. For OLMo-7B, the pretraining stages align with the following number of pretrained tokens: 177B, 500B, 1.5T. For OLMo-1B, the pretraining stages align with the following number of pretrained tokens: 168B, 500B, 1.5T.

| Model Size | Pretraining stage | | |
| --- | --- | --- | --- |
| | **Early** | **Mid** | **Late** |
| **OLMo-1B** | 0.000398 | 0.000379 | 0.000230 |
| **OLMo-7B** | 0.000280 | 0.000237 | 0.000101 |

# E   Additional Figures for the Pretraining Experiments

## E.1   Training dynamics of other OLMo-7B checkpoints

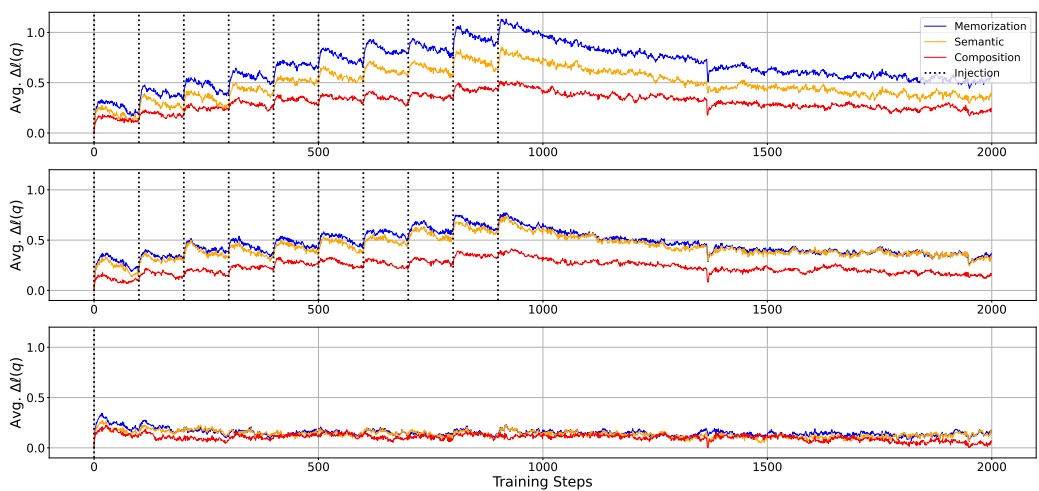

Figure 6: Training dynamics of OLMo-7B *Early* (170B) checkpoint.

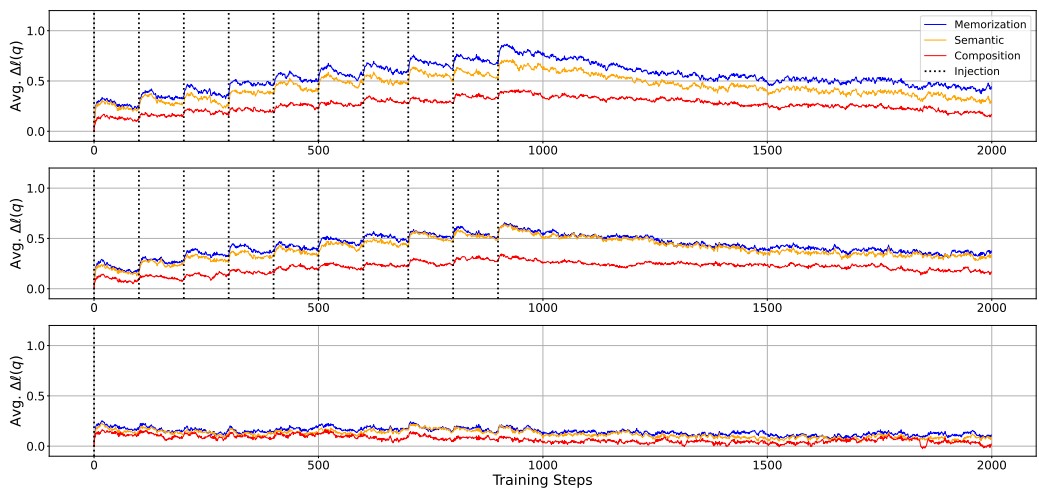

Figure 7: Training dynamics of OLMo-7B *Late* (1.5T) checkpoint.

## E.2 Training dynamics of other OLMo-1B checkpoints

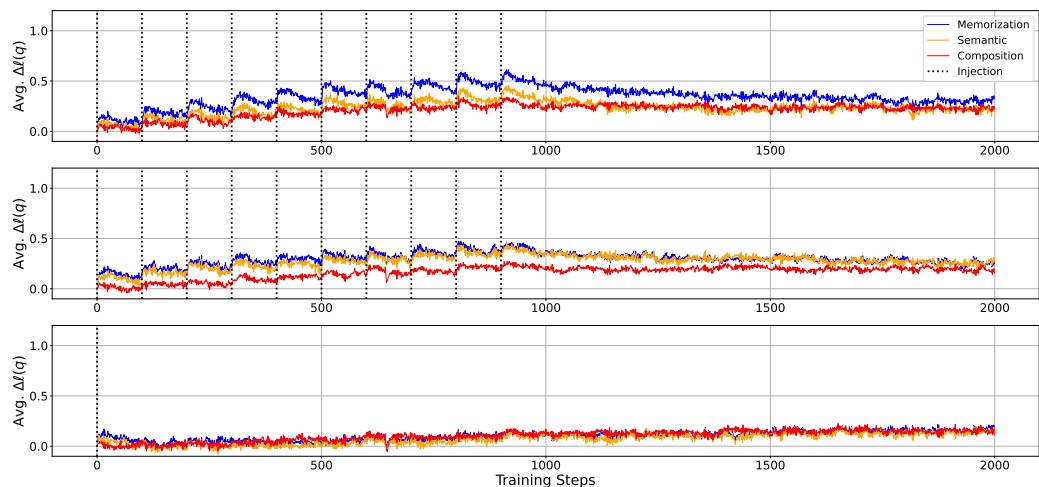

Figure 8: Training dynamics of OLMo-1B *Early* (170B) checkpoint. In comparison to the checkpoints of OLMo-7B and later checkpoints of OLMo-1B, the curves exhibit much more drastic fluctuations.

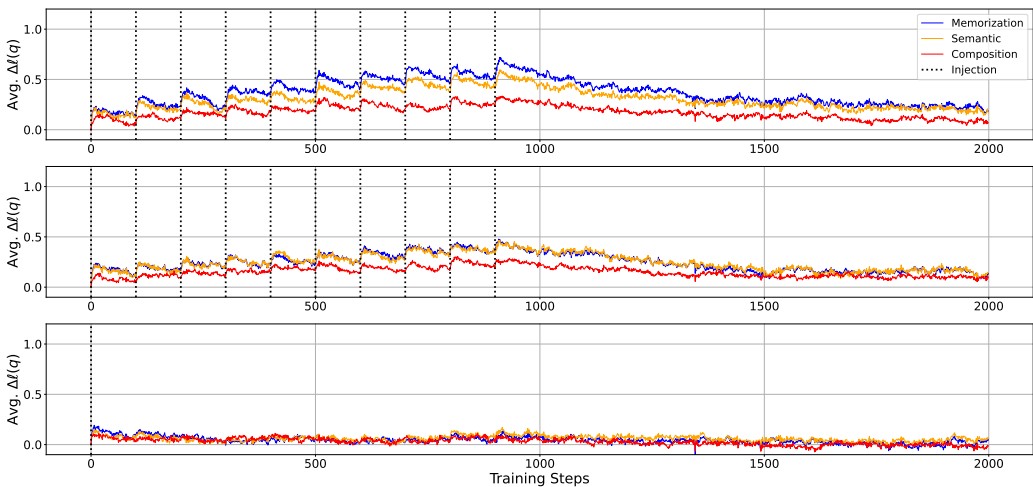

Figure 9: Training dynamics of OLMo-1B *Mid* (500B) checkpoint.

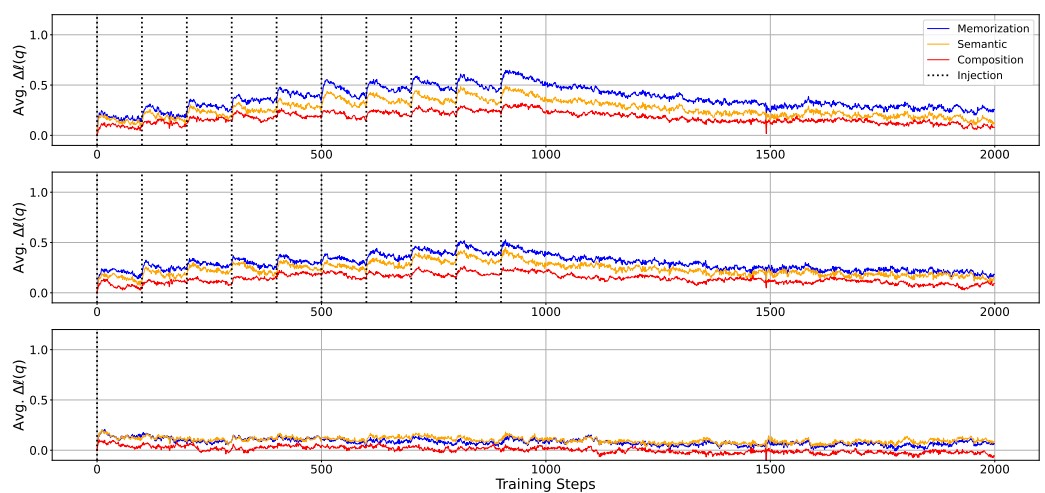

Figure 10: Training dynamics of OLMo-1B *Late* (1.5T) checkpoint.

## E.3 Effectivity measurement data for OLMo-1B

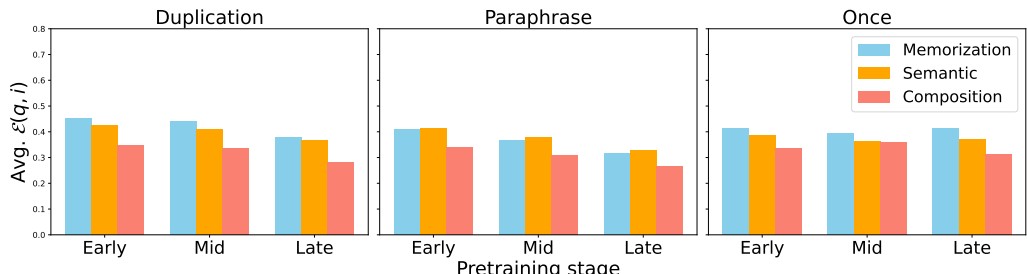

Figure 11: Effectivity measured for OLMo-1B models.

## E.4 Forgetting dynamics of OLMo-7B checkpoints

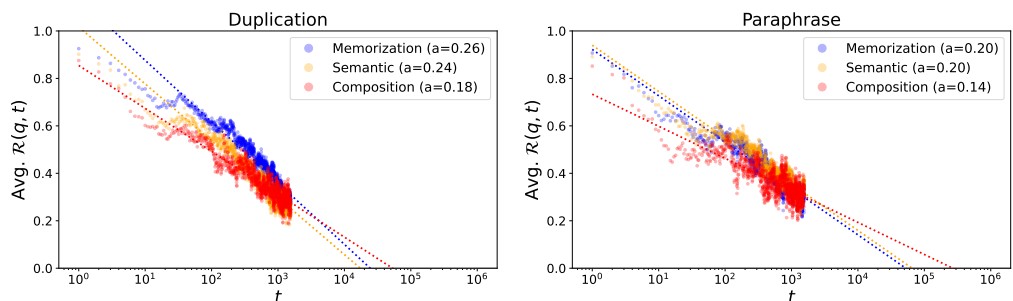

Figure 12: Forgetting dynamics of OLMo-7B *Early* (170B) checkpoint.

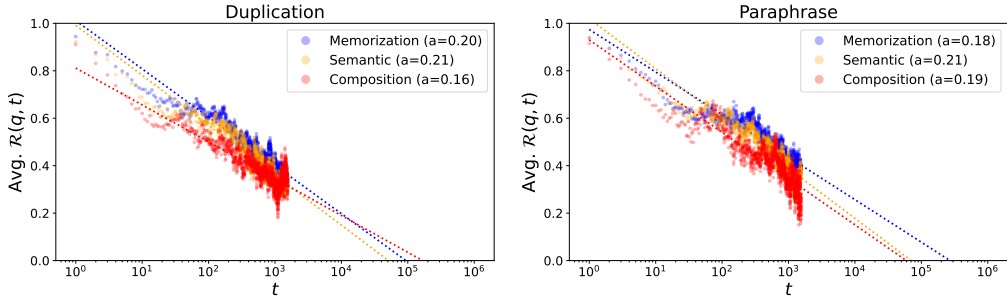

Figure 13: Forgetting dynamics of OLMo-7B *Late* (1.5T) checkpoint.

Table 6: Anticipated x-intercepts of $\mathcal{R}(p, t)$ measured with OLMo-7B, at three different pretraining stages, acquisition depths, and injection scenarios. The units are log(Tokens).

| Pretraining stage | | Early (170B) | Middle (500B) | Late (1.5T) |
|---|---|---|---|---|
| **Duplication** | Memorization | 11.01 | 11.02 | 11.59 |
| | Semantic | 10.86 | 10.98 | 11.33 |
| | Composition | 11.35 | 11.32 | 11.85 |
| **Paraphrase** | Memorization | 11.34 | 11.37 | 12.06 |
| | Semantic | 11.44 | 10.94 | 11.47 |
| | Composition | 12.05 | 11.88 | 11.40 |

## E.5 Forgetting dynamics of OLMo-1B checkpoints

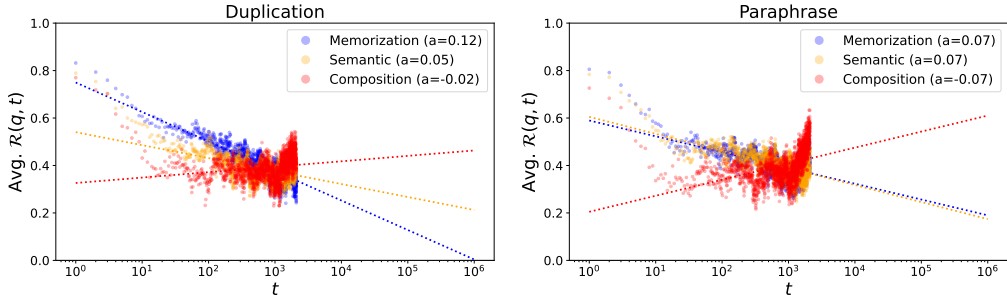

Figure 14: Forgetting dynamics of OLMo-1B *Early* (170B) checkpoint.

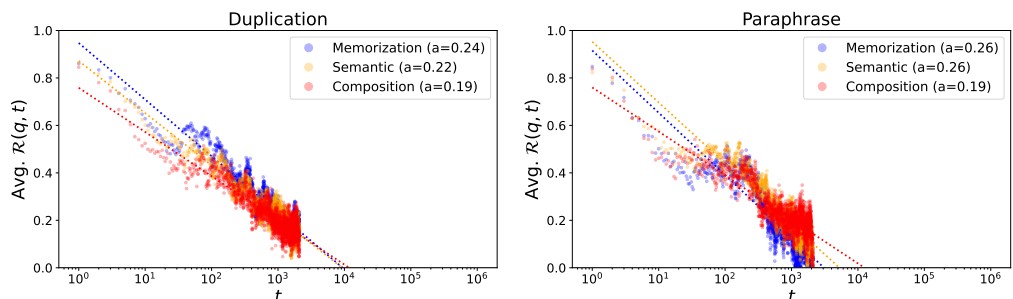

Figure 15: Forgetting dynamics of OLMo-1B *Mid* (500B) checkpoint.

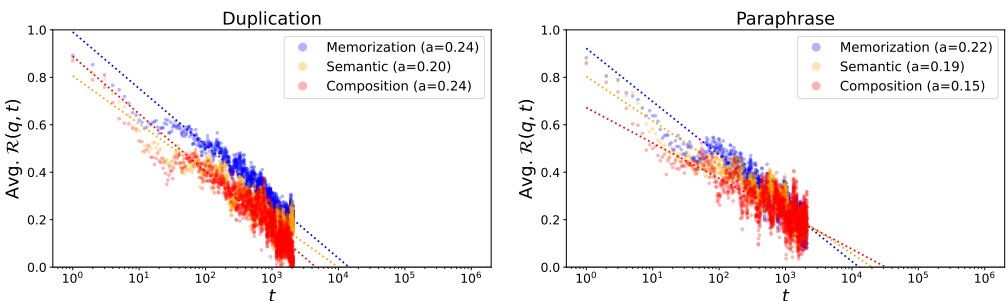

Figure 16: Forgetting dynamics of OLMo-1B *Late* (1.5T) checkpoint.

Table 7: Decay constant of average retainability ($\mathcal{R}(p,t)$) measured with OLMo-1B, at three different pretraining stages, acquisition depths, and injection scenarios. The values for the Early (168B) checkpoint are omitted due to the poor linear fitting ($R^2 < 0.4$), which is attributed to the highly unstable dynamics as shown in Appendix Figure 8 and 14.

| Pretraining Stage | | Early (168B) | Mid (494B) | Late (1.5T) |
|---|---|---|---|---|
| **Duplication** | Memorization | $0.12 \pm 0.0018$ | $0.24 \pm 0.0021$ | $0.24 \pm 0.0018$ |
| | Semantic | − | $0.22 \pm 0.0020$ | $0.20 \pm 0.0024$ |
| | Composition | − | $0.19 \pm 0.0021$ | $0.24 \pm 0.0026$ |
| **Paraphrase** | Memorization | − | $0.26 \pm 0.0031$ | $0.22 \pm 0.0021$ |
| | Semantic | − | $0.26 \pm 0.0024$ | $0.19 \pm 0.0022$ |
| | Composition | − | $0.19 \pm 0.0027$ | $0.15 \pm 0.0028$ |

Table 8: Anticipated x-intercepts of $\mathcal{R}(p,t)$ measured with OLMo-1B, at three different pretraining stages, acquisition depths, and injection scenarios. The units are log(Tokens). The values for the Early (168B) checkpoint are omitted due to the poor linear fitting ($R^2 < 0.4$), as mentioned in Appendix Table 7.

| Pretraining stage | | Early (168B) | Mid (494B) | Late (1.5T) |
|---|---|---|---|---|
| **Duplication** | Memorization | 12.65 | 10.60 | 10.78 |
| | Semantic | − | 10.59 | 10.62 |
| | Composition | − | 10.69 | 10.28 |
| **Paraphrase** | Memorization | − | 10.11 | 10.73 |
| | Semantic | − | 10.34 | 10.93 |
| | Composition | − | 10.72 | 11.13 |

# F   Experiments for Training Olmo-7B Checkpoints With a Constant Learning Rate

We continue training each OLMo-7B checkpoint with a constant learning rate, to compare the effectivity and retainability of each checkpoint while excluding the impact of different learning rates. Optimizer states are loaded to promote a warm start of continued training. Due to the restriction of computational resources, we reduce the batch size from 2048 to 128 for this experiment. The value of the constant learning rate is obtained by averaging the starting learning rates of three checkpoints. We do not apply learning rate decay for this experiment. All other training conditions not mentioned are identical to the main experiment. The results in Appendix Figure 17 demonstrate that there is no improvement of average effectivity in later checkpoints, although all models are trained with the same learning rate. This supports that the non-increasing effectivity in pretraining progress is not attributed to the learning rate decay. Similarly, there is no decrease in the decay constants for the later checkpoints (Appendix Table 9). Note that the figures in §F.1 demonstrate that reducing the batch size does not significantly change the model's behavior in accumulating log probability during factual knowledge acquisition.

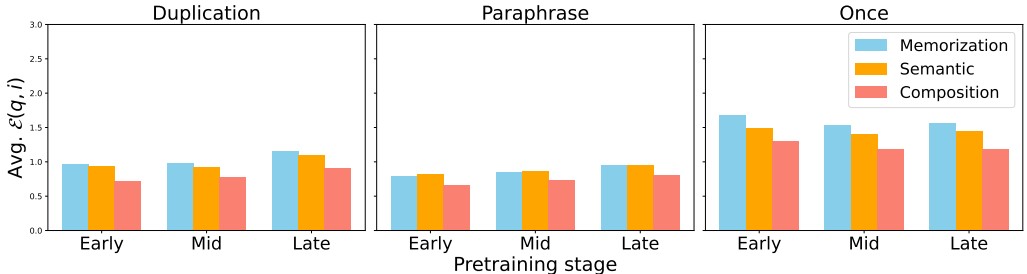

Figure 17: Average effectivity measured with OLMo-7B trained with a fixed constant learning rate.

Table 9: Decay constant of average retainability ($\mathcal{R}(p, t)$) measured with OLMo-7B trained with the same constant learning rate, at three different pretraining stages, acquisition depths, and injection scenarios. Note that the decay constant does not decrease for the later checkpoint.

| Pretraining stage | | Early (170B) | Mid (500B) | Late (1.5T) |
|---|---|---|---|---|
| **Duplication** | Memorization | $0.29 \pm 0.0017$ | $0.30 \pm 0.0025$ | $0.35 \pm 0.0025$ |
| | Semantic | $0.28 \pm 0.0015$ | $0.28 \pm 0.0023$ | $0.30 \pm 0.0020$ |
| | Composition | $0.28 \pm 0.0019$ | $0.28 \pm 0.0031$ | $0.25 \pm 0.0029$ |
| **Paraphrase** | Memorization | $0.29 \pm 0.0019$ | $0.31 \pm 0.0030$ | $0.33 \pm 0.0023$ |
| | Semantic | $0.30 \pm 0.0019$ | $0.30 \pm 0.0027$ | $0.32 \pm 0.0022$ |
| | Composition | $0.30 \pm 0.0022$ | $0.27 \pm 0.0031$ | $0.22 \pm 0.0034$ |

## F.1   Training dynamics for constant learning rate experiments

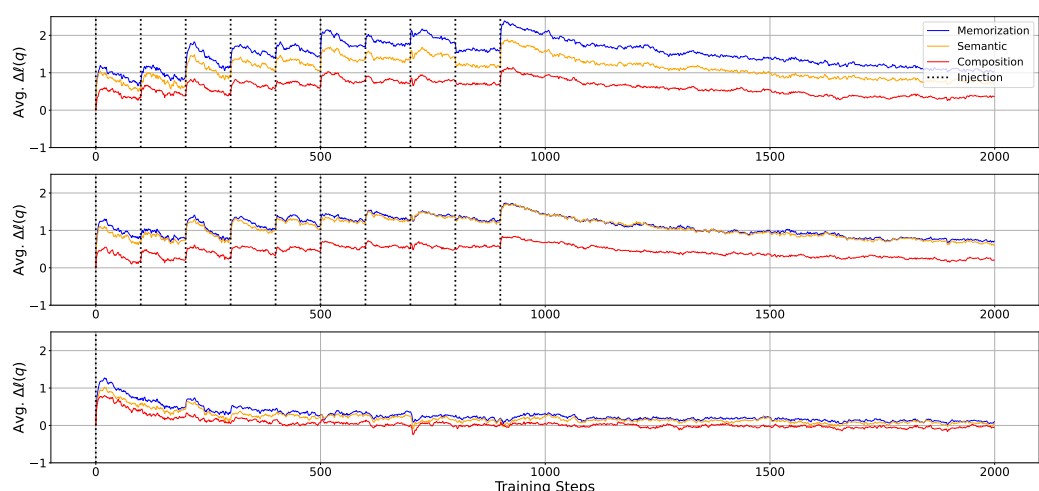

Figure 18: Training dynamics of OLMo-7B *Early* (170B) checkpoint trained with a constant learning rate.

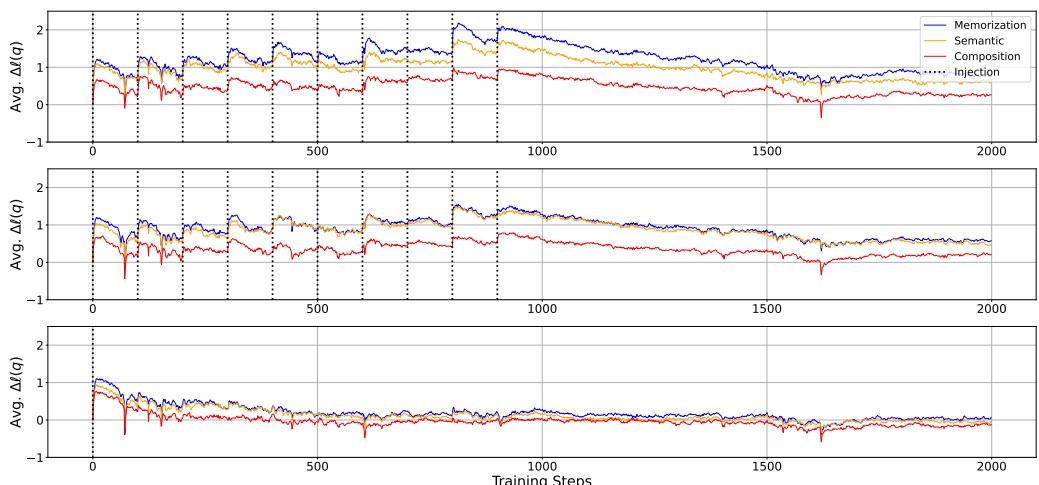

Figure 19: Training dynamics of OLMo-7B *Mid* (500B) checkpoint trained with a constant learning rate.

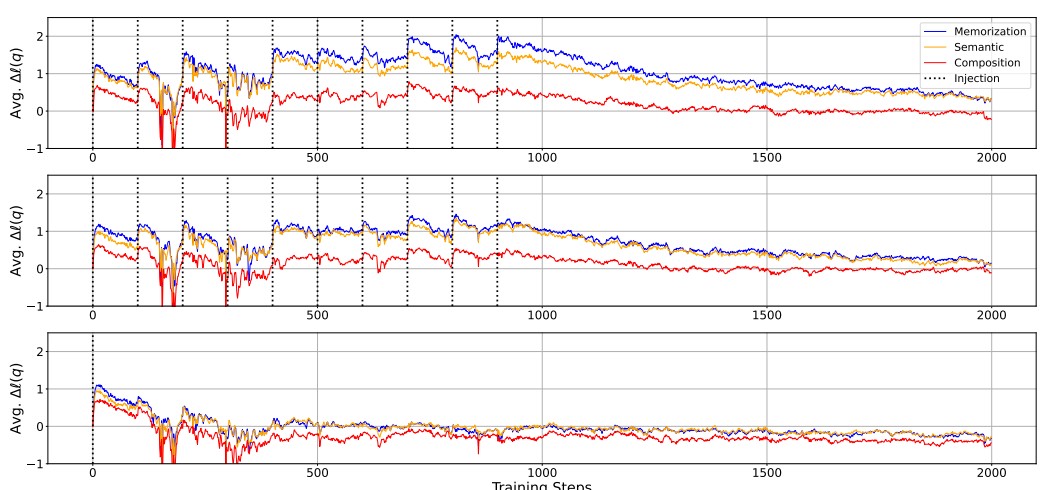

Figure 20: Training dynamics of OLMo-7B *Late* (1.5T) checkpoint trained with a constant learning rate.

# G    Forgetting Dynamics of Olmo-7B Trained With a Reduced Batch Size

Similar to F, we train the OLMo-7B intermediate checkpoints with a reduced batch size of 128. However, we set the learning rate for each checkpoint as the initial learning rate (Appendix Table 5), as the objective of this experiment is to examine the effect of reduced batch size on the forgetting dynamics. We re-initialize the optimizer state. We observe that this results in unstable dynamics in early steps, but the dynamics are stabilized soon, and do not harm the model's overall behavior in general (§G). Appendix Figure 21 shows the effectivity measurements of OLMo-7B models at different pretraining stages. Similar to the observations in Appendix Figure 17, the effectivity values are greater compared to the values in the pretraining experiment (Figure 3). Appendix Figure 22 and 23 illustrates the forgetting dynamics of OLMo-7B *Early* (170B) and *late* (1.5T) checkpoints, respectively. Appendix Table 10 shows the decay constants ($a$) measured with three different pretraining stages, acquisition depths, and injection scenarios. Note that the slope remains unchanged regardless of whether we set the x-axis to tokens or training steps. Hence, the decay constants in the table can be directly compared to the values presented in Table 2. Comparing the values of the expected x-intercepts of retainability presented in Appendix Table 11 with Appendix Table 6, the results demonstrate that the model trained with a smaller batch size has a shorter *learnability threshold*.

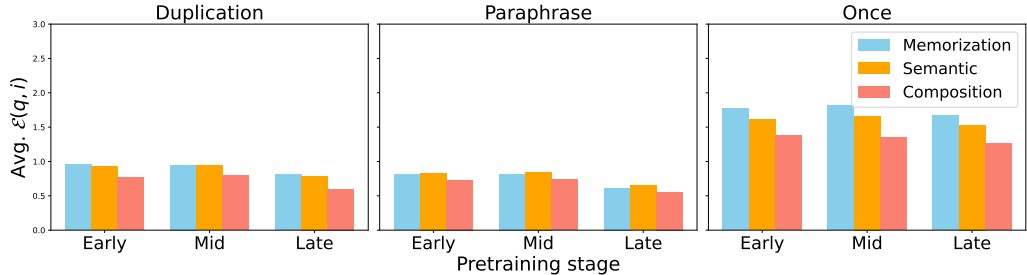

Figure 21: Average effectivity measured with OLMo-7B trained with a batch size of 128. The low effectivity values observed in the *once* injection scenario are attributed to the unstable dynamics after the re-initialization of the optimizer states.

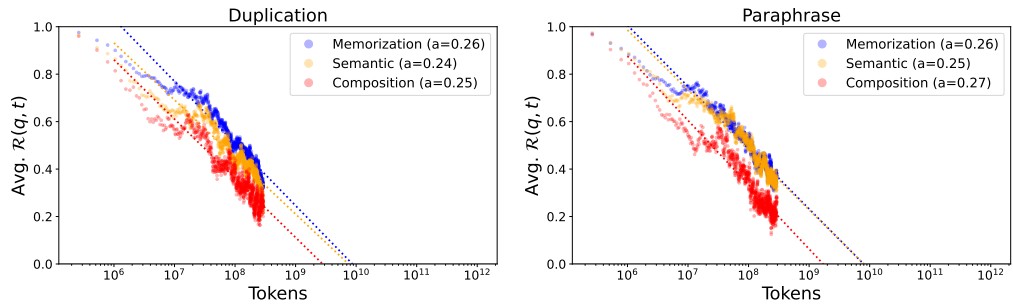

Figure 22: Forgetting dynamics of OLMo-7B *Early* (170B) checkpoint with a reduced batch size.

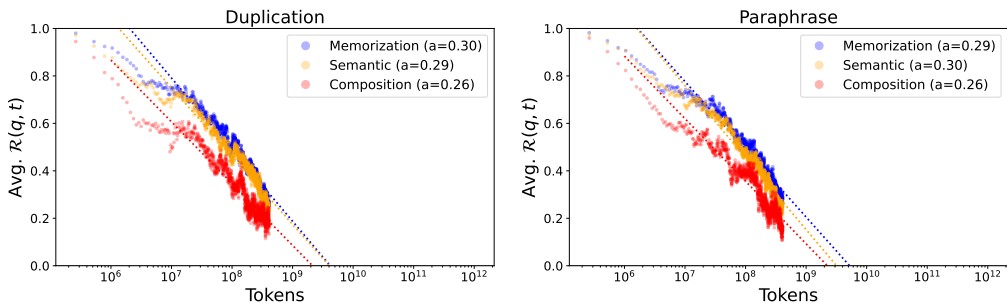

Figure 23: Forgetting dynamics of OLMo-7B *Late* (1.5T) checkpoint with a reduced batch size.

Table 10: Decay constant of average retainability ($\mathcal{R}(p, t)$) measured with OLMo-7B trained with a batch size of 128, at three different pretraining stages, acquisition depths, and injection scenarios.

| Pretraining stage | | Early (170B) | Mid (500B) | Late (1.5T) |
|---|---|---|---|---|
| **Duplication** | Memorization | $0.26 \pm 0.0024$ | $0.31 \pm 0.0021$ | $0.30 \pm 0.0022$ |
| | Semantic | $0.24 \pm 0.0027$ | $0.29 \pm 0.0019$ | $0.29 \pm 0.0022$ |
| | Composition | $0.25 \pm 0.0027$ | $0.26 \pm 0.0018$ | $0.26 \pm 0.0021$ |
| **Paraphrase** | Memorization | $0.26 \pm 0.0022$ | $0.31 \pm 0.0020$ | $0.29 \pm 0.0020$ |
| | Semantic | $0.25 \pm 0.0025$ | $0.32 \pm 0.0026$ | $0.30 \pm 0.0021$ |
| | Composition | $0.27 \pm 0.0028$ | $0.26 \pm 0.0024$ | $0.26 \pm 0.0023$ |

Table 11: Anticipated x-intercepts of $\mathcal{R}(p, t)$ measured with OLMo-7B trained with a batch size of 128, at three different pretraining stages, acquisition depths, and injection scenarios. The units are log(Tokens).

| Pretraining stage | | Early (170B) | Mid (500B) | Late (1.5T) |
|---|---|---|---|---|
| **Duplication** | Memorization | 9.94 | 9.45 | 9.62 |
| | Semantic | 9.87 | 9.49 | 9.61 |
| | Composition | 9.45 | 9.47 | 9.33 |
| **Paraphrase** | Memorization | 9.90 | 9.44 | 9.72 |
| | Semantic | 9.90 | 9.39 | 9.50 |
| | Composition | 9.23 | 9.28 | 9.35 |

## G.1 Training dynamics for experiments on the forgetting dynamics with a reduced batch size

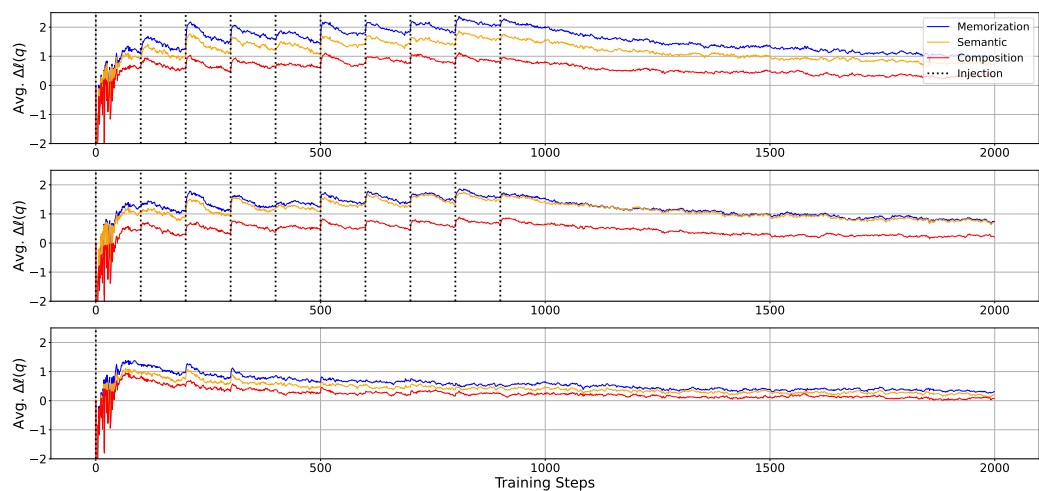

Figure 24: Training dynamics of OLMo-7B *Early* (170B) checkpoint trained with reduced batch size and re-initialized optimizer state.

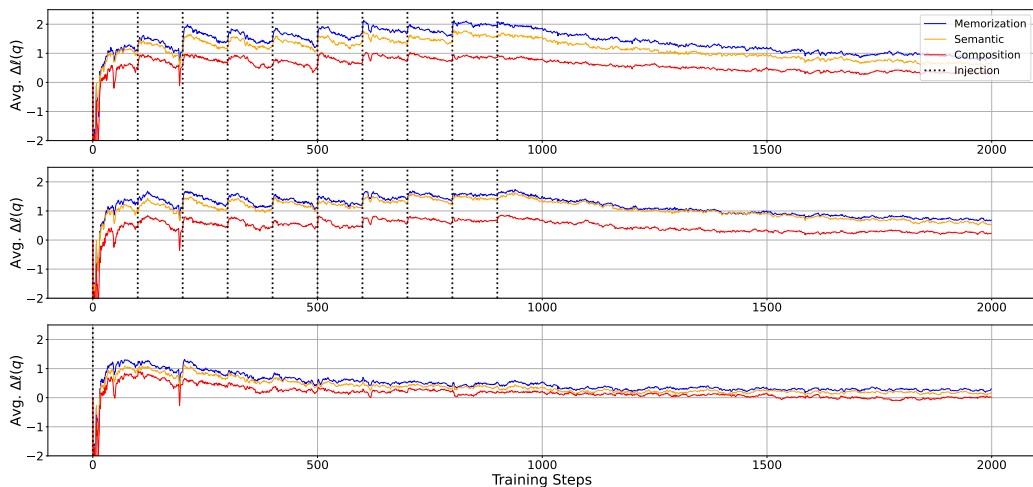

Figure 25: Training dynamics of OLMo-7B *Mid* (500B) checkpoint trained with reduced batch size and re-initialized optimizer state.

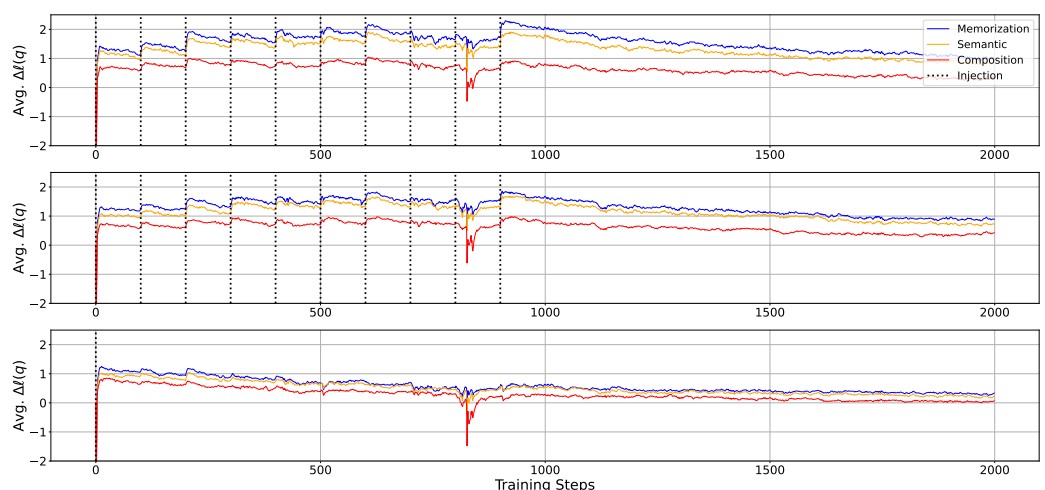

Figure 26: Training dynamics of OLMo-7B *Late* (1.5T) checkpoint trained with reduced batch size and re-initialized optimizer state.

# H Effect of the Number of Previous Encounters on Effectivity and Retainability of Factual Knowledge

We measure the average effectivity for each count of injection ($i$) in *duplication* and *paraphrase* injection scenario. In this analysis, we exclude the cases where the log probability at the local acquisition maxima is smaller than the point before the model is trained with the injected knowledge, as such cases can be regarded as failure cases of learning. Appendix Figure 27, 28, and 29 display the results for OLMo-7B *early*, *mid*, and *late* checkpoints, respectively. We observe that the effectivity is relatively constant regardless of the number of previous injections of the knowledge. However, we observe that the effectivity is the highest when the model is trained with the injected knowledge for the first time, both in the *duplication* and *paraphrase* injection scenarios.

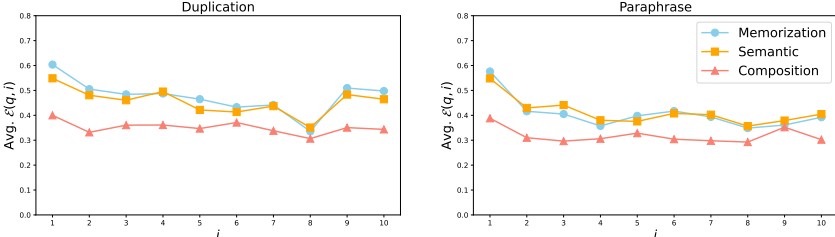

Figure 27: Average effectivity measured for each count of injection, measured with OLMo-7B *Early* (170B) checkpoint.

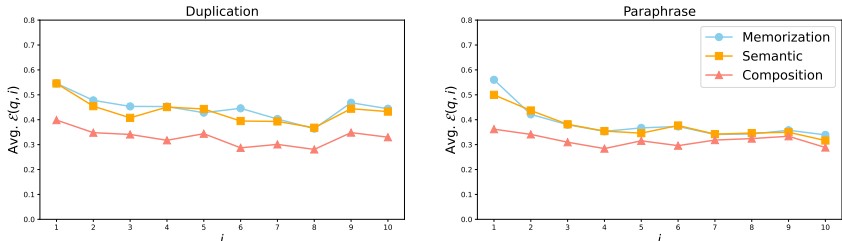

Figure 28: Average effectivity measured for each count of injection, measured with OLMo-7B *Mid* (500B) checkpoint.

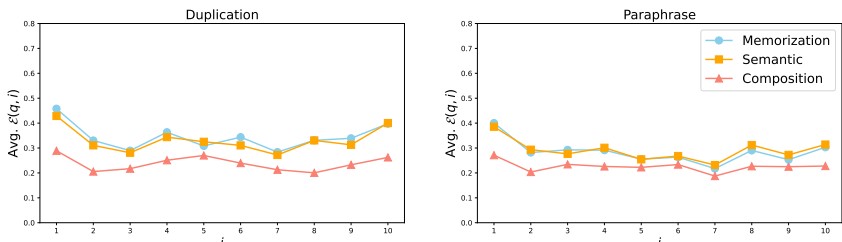

Figure 29: Average effectivity measured for each count of injection, measured with OLMo-7B *Late* (1.5T) checkpoint.

