# OpenReview forum: "How Do Large Language Models Acquire Factual Knowledge During Pretraining?"
_NeurIPS.cc/2024/Conference — NeurIPS 2024 poster_

### Official Review · Reviewer_1qMb · 2024-06-30

**Soundness:** 3
**Presentation:** 2
**Contribution:** 2
**Rating:** 5
**Confidence:** 3

**Summary:**

This paper aims at understanding how large language models acquire knowledge from the pretraining process.
To this end, the authors propose a dataset consisting of paragraphs about fictional entities and three different kinds of probes that can be used to test whether the model successfully acquires the knowledge after fine-tuning.
Their finding includes:

- The likelihood of the knowledge gradually increases as the models are exposed to the knowledge more times, but the models also gradually forget the knowledge as the models are fine-tuned with other data.
- Larger models acquire knowledge more effectively, but models trained with more tokens do not.
- A larger batch size helps models to retain the knowledge better (forget slower).

They suggest that this explains why models can not learn long-tail knowledge.

**Strengths:**

- The authors conduct comprehensive investigations on LMs' knowledge acquisition from different aspects, including
    - Types of knowledge recurrence (once/duplication/paraphrase)
    - Model scales (1B/7B)
    - Different batch size (16/4096)
    - Different level of acquisition (memorization/semantic/compositional probs)
- Figure 2 shows how knowledge is accumulated in the model throughout the training process.
- Their results explain why language models cannot acquire long-tail knowledge well.
- The results about batch size are interesting and may be useful for choosing the best batch size in practice.

**Weaknesses:**

1. The interpretation about batch size may be too assertive. It is not certain that the retention rate will really go to 0 as the number of tokens increases. The interpretation is based on extrapolation.
2. Also, it is possible that when the batch size is larger than a certain number, the model will not be able to learn long-tail knowledge in the batch. The authors should also plot the relationship between batch size and effectivity.
2. The evaluation metric (measuring the likelihood of the target sequence) may not effectively reflect whether the model acquires the knowledge, as the target phrase may not be the only correct continuation (at least it is unclear in the main text).
3. Line 70: [13] does not study NLP tasks!
4. The writing could be improved so the information is more digestible. For example, the first paragraph in Section 4.2 contains information about model scales and pretraining stages. It's better to be re-organized as two paragraphs.
5. In Section 3, the authors should provide more details about how the dataset is created. Also, the presentation could be greatly improved. The purpose of the experiment, the purpose of the dataset, the design of the dataset, the way they generate the dataset, the inspection of the examples, should be in separated paragraphs.
5. This paper defines many symbols but only few of them are used frequently throughout the whole paper. Meanwhile, $\delta \ell (q)$ is not defined in the main text, but appears in two figures. I suggest the authors simplify the narrative to improve the readability of this paper.

I acknowledge the contribution of this paper. However, the presentation induces unnecessary difficulty to understand and assess this work. I suggest that this work needs more polish before being published.

**Questions:**

1. $\theta_t$ is the parameter before the $t$-th update?
2. In Figure 1, why can the number of training steps be minus?
3. In convention, $\mathcal{E}$ is usually used for error. The authors could consider using another symbol for effectivity.

**Limitations:**

Limitations are addressed in Appendix A.

---

> ### Author Rebuttal · Authors · 2024-08-06
>
> We sincerely thank reviewer 1qMb for the detailed review and valuable feedback.
>
> We will respond to the weaknesses and questions in order:
>
> > The interpretation about batch size may be too assertive. It is not certain that the retention rate will really go to 0 as the number of tokens increases. The interpretation is based on extrapolation.
>
> We acknowledge the need for extended training to minimize extrapolation in Figures 4 and 5. However, given the significant computational burden of pretraining experiments (L690-691), such extensions are currently infeasible in our academic setting. We aim to address this with additional resources in future work.
> Importantly, Figure 16 (Appendix E.5) demonstrates the power-law holds near the x-axis for 1B models, which show faster forgetting than 7B models, supporting our interpretation's reliability.
>
> > The authors should also plot the relationship between batch size and effectivity.
>
> The effect of reducing the batch size on effectivity is demonstrated in Figure 21 in Appendix G (Note: the LR was fixed for the experiments with the reduced batch size, and L246-248 &  L725-727 are corrected in our latest version). Comparing Figure 2 with Figure 21, our results suggest that effectivity is increased with reduced batch size.
>
> > it is possible that when the batch size is larger than a certain number, the model will not be able to learn long-tail knowledge in the batch.
>
> We acknowledge the need for experiments with various batch sizes (noted in Limitations, L504-506) to generalize to extreme values. However, our experiments with batch sizes used in actual OLMo pretraining (4M tokens) provide significant implications for recent LLMs in practice.
>
> > The evaluation metric (measuring the likelihood of the target sequence) may not effectively reflect whether the model acquires the knowledge, as the target phrase may not be the only correct continuation (at least it is unclear in the main text).
>
> This is a good point! While the target span may not be the only correct continuation, we believe this does not have a significant impact on our interpretations:
>
> - Knowledge acquisition should increase log probability for all correct continuations, including our target span.
> - Using fictitious and factual knowledge mitigates the effects of multiple possible answers.
> - Our focus is on relative comparisons across training conditions, not absolute likelihood increments.
>
> These factors mitigate concerns about multiple-answer scenarios affecting our conclusions.
>
> > Line 70: [13] does not study NLP tasks!
>
> Thank you for pointing this out. We will correct it.
>
> > the first paragraph in Section 4.2 contains information about model scales and pretraining stages. It's better to be re-organized as two paragraphs.
>
> > The purpose of the experiment, the purpose of the dataset, the design of the dataset, the way they generate the dataset, the inspection of the examples, should be in separated paragraphs.
>
> We appreciate your thoughtful suggestions to make our paper more readable. We will improve them in the revised version.
>
> > In Section 3, the authors should provide more details about how the dataset is created.
>
> Although we could not provide all the details in the main text due to the page limit, the detailed dataset generation process is in Appendix B (referenced in L98-99):
>
> - Generation and filtering process (L509-535)
> - Full examples in Tables 3 and 4
> - Exact prompts used in Appendix C
>
> We'll add more details to the main text in the revised version.
>
> > This paper defines many symbols but only few of them are used frequently throughout the whole paper.
>
> Our symbol definitions have dual purposes: quantifying our experimental results and providing a framework for future studies on LLM pretraining dynamics. While this may slightly compromise readability, it ensures interchangeability between studies with different setups. In the revised version, we will enhance the clarity and simplicity of our metrics and symbols, enabling readers to easily grasp their meanings while maintaining adaptability. However, we believe that oversimplifying definitions could limit their applicability in future work. For example, defining $t_{LAM}(q,i)$ and $\mathcal{E}(q,i)$ without considering $i$ (the $i$-th encounter of knowledge) would make measured values incompatible with future studies using different experimental setups, such as varying injection intervals.
>
> > 𝛿ℓ(𝑞)  is not defined in the main text, but appears in two figures.
>
> $\Deltaℓ(q)$ is explained in Figure 1's caption where it's first introduced. We will add its definition to the main text as well.
>
> > 𝜃𝑡 is the parameter before the 𝑡-th update?
>
> Yes, we will clarify this in the revised version.
>
> > In Figure 1, why can the number of training steps be minus?
>
> In Figure 1, negative training steps represent the period before the model encounters the injected knowledge. We set $t=0$ as the reference point where the model is exposed to the minibatch containing the injected knowledge. For $t\leq0$, the model is updated only with the Dolma corpus.
>
> > In convention, 𝐸 is usually used for error. The authors could consider using another symbol for effectivity.
>
> Thank you for the feedback. We will use an alternative symbol in the revised version.
>
> We hope our responses have addressed your concerns. We appreciate your thorough review and remain open to any further questions.

---

> > ### Comment · Reviewer_1qMb · 2024-08-07
> >
> > Thanks for the clarifications. I am happy to increase my score by 1.

---

### Official Review · Reviewer_fwEj · 2024-07-02

**Soundness:** 2
**Presentation:** 3
**Contribution:** 2
**Rating:** 4
**Confidence:** 4

**Summary:**

The paper proposes to inject synthetic factual knowledge in the pretraining data of large language models and measure the acquisition and retention of this knowledge over time. The study shows that model acquires factual knowledge gradually with many exposures of the same fact, and forgets over time if not reinforced. The results also show that forgetting follows a power-law relationship with training steps, and deduplication of training data can improve knowledge retention.

**Strengths:**

- The paper uses a novel approach to study the acquisition and retention of factual knowledge in the pretraining phase of language models, by inserting synthetic facts into the training data and measuring the model's recall of these facts along the training process.
- The experimental design is thorough, covering different model sizes, batch sizes, and pretraining stages.
- The paper is well-structured and clearly written, with detailed explanations of the methodology and results.
- The work provides insights into why LLMs struggle with long-tail knowledge and how deduplication can enhance model performance, which are critical considerations for developing better strategies to improve LM training.

**Weaknesses:**

- Counterintuitive results not explained in the main experiment: In figure 2, the model trained with paraphrased knowledge show lower performance after learning on the semantic probe and the composition probe compared to the model trained with duplicated knowledge. This is counterintuitive as more variations in the training data should lead to better generalization, and semantic probe and composition probe are designed to measure generalization. (The author seems wanting to explain this in line 212-215, but not arriving at an explanation in the end: duplicated knowledge forgets faster, but why it also learns faster?)
- An important factor that controls learning and forgetting, the learning rate, is not explicitly considered in the experiments. The lack of analysis on the impact of learning rate leaves many conclusions open to alternative interpretation. For example, could the gradual learning of facts be an artifact of low learning rate? If the learning rate is raised, could the model learn in fewer (or even a single) steps? For the power-law relationship between forgetting and training steps, the decay rate is also likely affected by the learning rate. The authors should investigate the impact of learning rate on the learning and forgetting dynamics to provide a more comprehensive understanding of the observed phenomena.
- The discussion on data deduplication seems contradictory to the results and the "learnability threshold": the author claims that deduplication slows down forgetting as is observed in Figure 2, but Figure 2 shows that deduplication also slows down learning. How to tradeoff between faster learning and slower forgetting is not obvious from the results (at the end of the training session in Figure 2, duplicated knowledge still has a higher performance than deduplicated knowledge). Also, deduplication increases the average interval between exposures, which could cause more knowledge to fall outside the learnability threshold and thus adversely affect knowledge learning. This is another contradiction that needs to be addressed.

Below are somewhat minor issues, but still affecting the credibility of the results:
- The training and evaluation data is not described clearly enough in main text: the authors should provide more description of the generated fictional knowledge paragraphs, for example, how it is generated, the main statistics of the passages, how is the "target span" selected, etc. The evaluation probes are also not specified in sufficient detail, for example, how is the composition probe constructed, and what does "composing factual knowledge in multiple sentences" exactly mean? These details are crucial for appreciating the experimental results and need to be included in the main text.
- Some important experiment details are missing: in line 156, what does the "paraphrased knowledge" refer to? Is it the same as the passages in semantic probe? How many paraphrases are used for each fact? These details could significantly affect the generalization of the learned knowledge and should be clarified.
- The idea of "learnability threshold" is cherishable, but may not be reliable enough as an explanation: although the observed forgetting fits to a power-law curve, extra care should be taken when extrapolating the curve to extreme values, such as to the vicinity of the x-axis, where the curve is likely to diverge from the power-law (or change to another piecewise power-law). More exact results are needed to measure the learnability threshold instead of extrapolating from far-away.
- The relationship between model size and forgetting is not sufficiently explored: it is often speculated that larger models may forget slower due to their larger capacity. Would the methodology used in this paper be able to give a precise characterization of the relationship between model size and forgetting?

**Questions:**

Please refer to the above section.

**Limitations:**

The authors adequately addressed the limitations.

---

> ### Author Rebuttal · Authors · 2024-08-06
>
> We sincerely thank reviewer fwEj for the detailed review and valuable feedback. We will respond to each weakness and question in order:
>
> > the model trained with paraphrased knowledge show lower performance after learning on the semantic probe and the composition probe compared to the model trained with duplicated knowledge. This is counterintuitive as more variations in the training data should lead to better generalization
>
> This point stems from a misunderstanding between overall model performance (i.e., generalization) and rapid acquisition of specific knowledge (measured by effectivity). **Our analysis focused on the gap between memorization and generalization improvements, not absolute increases. We didn't interpret immediate improvement magnitude as an indicator of model performance.**
> While duplication shows higher effectivity, it leads to a widening memorization-generalization gap over iterations. This results in the model favoring memorized content, potentially harming generalization (L295-299). Previous research has shown that LLMs trained on duplicated datasets tend to generate memorized content, which can be mitigated through deduplication [1].
>
> > but why it also learns faster?
>
> The slightly larger effectivity for duplication likely stems from the greater lexical overlap between injected knowledge and probes, as we used injected knowledge as probe templates.
>
> > the learning rate, is not explicitly considered in the experiments.
>
> Agreed, as noted in Limitations (L504-506). Given the computational intensity of pretraining experiments (L690-691), exploring learning rate effects is currently infeasible in our academic setting. We acknowledge the need for further investigation with additional resources.
>
> > could the gradual learning of facts be an artifact of low learning rate?
>
> Experimental data with an unusually high learning rate (10e-3, see attached Figure 1) demonstrate that the accumulating behavior remains consistent, indicating it's not an artifact of a low learning rate.
>
> > How to tradeoff between faster learning and slower forgetting is not obvious from the results (at the end of the training session in Figure 2, duplicated knowledge still has a higher performance than deduplicated knowledge).
>
> We don't interpret absolute log probability increases as performance measures. For instance, higher probabilities for target spans in memorization probes in the duplication scenario (Figure 2) don't necessarily indicate better performance. We claim that deduplication improves overall performance by mitigating memorization-generalization gaps, given sufficient training with deduplicated data.
>
> > Also, deduplication increases the average interval between exposures, which could cause more knowledge to fall outside the learnability threshold
>
> Deduplication can actually bring more knowledge within the learnability threshold. We might first consider a synthetic case in which every instance is tripled. In this scenario, deduplication does not alter the expected exposure interval, as the dataset size decreases proportionally. In real pretraining corpora, most instances aren't duplicated, while few are highly duplicated ([1] showed this with the C4 dataset). Thus, deduplication typically reduces the expected interval for most knowledge, and deduplication can bring more knowledge inside the learnability threshold.
>
> > the authors should provide more description of the generated fictional knowledge
>
> We provided main statistics in L92-93 and addressed the detailed dataset generation process and target span selection in Appendix B (referenced in L98-99). Space constraints limited main text details, but we'll clarify the dataset design further in the revised version.
>
> > how is the composition probe constructed, and what does "composing factual knowledge in multiple sentences" exactly mean?
>
> We defined compositional generalization as “the ability to understand and combine the implications of different factual knowledge presented in a passage and apply them to deduce unseen knowledge”. Composition probes were created based on this definition, as detailed in Appendix C.4 (L577-59). We will clarify this in the main text.
>
> > “Some important experiment details are missing: in line 156, what does the "paraphrased knowledge" refer to?
>
> This is addressed in Appendix B. “paraphrased knowledge” is constructed by prompting GPT-4 to paraphrase each injected knowledge (L517-518), using the prompt in Appendix C.2 (L553-556).
>
> > Is it the same as the passages in semantic probe?
>
> No. Semantic probes are separate GPT-4 paraphrases of each memorization probe sentence, maintaining the target span (Appendix B, L523-525). The specific prompt is in Appendix C.3 (L557-576).
>
> > How many paraphrases are used for each fact?
>
> We used 9 paraphrases per injected knowledge (10 total injections: 1 original + 9 paraphrased), as described in Appendix B (L517-518).
>
> > although the observed forgetting fits to a power-law curve, extra care should be taken when extrapolating the curve to extreme values, such as to the vicinity of the x-axis
>
> We agree caution is needed in extrapolation. However, Figure 16 (Appendix E.5) demonstrates the power-law holds near the x-axis for 1B models, which show faster forgetting than 7B models, supporting our extrapolation's reliability.
>
> > The relationship between model size and forgetting is not sufficiently explored
>
> We explored this through comparisons of decay constants for 7B models (Table 2, Section 4.3) and 1B models (Table 7, Appendix E.4). Results show larger models are more robust to forgetting, aligning with previous findings.
>
> We thank Reviewer fwEj again for their time and effort. We welcome any follow-up questions and are open to further discussion.
>
> ### References
> [1] https://arxiv.org/abs/2107.06499

---

> > ### Comment · Reviewer_fwEj · 2024-08-10
> >
> > Thank you for your response, which provided sufficient information that clarified many of my questions. But for my main concern (interpretation of the results), I don't find the authors' response convincing. The metric of the paper (e.g., effectivity) are defined with log probability l, and I don't think these metrics can be meaningfully interpreted without the log probability themselves being interpretable and reflect model performance. Or did the authors find any confounding factors that affect the log probability values but can be removed by subtracting two log probabilities?
> >
> > Based on this, I have raised my score by 1 but still maintain some concern about the validity of the methodology.

---

> > > ### Author Response · Authors · 2024-08-12
> > >
> > > We appreciate your acknowledgment of our rebuttal, the subsequent score increase, and your thoughtful follow-up questions. We welcome the opportunity to elaborate further on the intuition behind the log probability-based evaluation in our work.
> > >
> > > Perplexity, a metric directly related to log probability, has been conventionally adopted as a performance measure for language modeling in NLP literature. However, recent work has questioned whether ‘lower is always better’ holds for perplexity, since low perplexity does not necessarily imply that the model represents the true distribution of natural language [1], nor does it guarantee high performance [2]. Therefore, we believe that we cannot conclude that a model ‘knows better’ about a given knowledge solely based on the fact that it provides a higher probability to the probe sequence.
> > >
> > > In that sense, our interpretation of the log probability is drier; a model providing an increased log probability to a given sequence indicates a higher likelihood that the sequence is generated during model inference. This can manifest as an improvement in inference-based metrics (e.g., accuracy) at some point.
> > > While this may seem somewhat trivial, this is precisely what we aimed to investigate. Although log probability itself may not directly represent performance, it can provide fine-grained information as a progress measure on how knowledge will be ‘revealed’ in generated sequences (This interpretation is closely aligned with [3]).
> > >
> > > Unfortunately, we cannot determine the exact log probability value at which the model begins to generate factual knowledge, or where it might be considered too high and enter an 'overconfident' region, as this is highly context-dependent. To address this, we focused on measuring the relative increase in log probability after the model encounters previously unseen knowledge. This approach helps mitigate potential overconfidence or interference issues, as we're dealing with novel information for the model. By comparing log probabilities before and after exposure, we can isolate the effect of encountering specific knowledge and gain insights into how the model incorporates new factual information during pretraining.
> > >
> > > We appreciate your engagement with our work and remain open to further discussion.
> > >
> > > ## References
> > >
> > > [1] https://arxiv.org/abs/1904.09751
> > >
> > > [2] https://arxiv.org/abs/2109.09115
> > >
> > > [3] https://arxiv.org/abs/2304.15004

---

### Official Review · Reviewer_yByP · 2024-07-09

**Soundness:** 3
**Presentation:** 3
**Contribution:** 3
**Rating:** 7
**Confidence:** 3

**Summary:**

This paper explores the process by which LLMs accumulate factual knowledge during pretraining. It finds that while more data exposure can improve immediate knowledge acquisition, it does not significantly affect long-term retention due to subsequent forgetting. The study reveals that larger batch sizes and deduplicated training data enhance knowledge retention, and it identifies a power-law relationship between training steps and forgetting rates. The paper introduces a FICTIONAL KNOWLEDGE dataset and metrics to analyze knowledge acquisition dynamics, providing insights into the challenges LLMs face with long-tail knowledge and the benefits of data deduplication. These findings contribute to a more nuanced understanding of LLM pretraining and have implications for improving model reliability and performance.

**Strengths:**

- Useful Resource: The creation of the FICTIONAL KNOWLEDGE dataset allows for controlled experiments to simulate and analyze the acquisition of new knowledge by LLMs.
- Evaluation Framework: The paper provides a detailed examination of knowledge acquisition at different depths—memorization, semantic generalization, and compositional generalization—offering a nuanced understanding of how LLMs process and retain information. The introduction of metrics like local acquisition maxima, effectivity, and retainability provides a quantitative framework to assess the dynamics of knowledge acquisition and forgetting in LLMs.
- Empirical Analysis:  The study takes a comprehensive approach by considering various factors that influence knowledge acquisition, including model scale, pretraining stage, and the nature of the training data. The empirical evidence presented challenges common assumptions, such as the belief that more data always leads to better knowledge retention, and highlights the importance of training conditions like batch size and data deduplication.
- Insights: The research offers practical insights into improving LLM training, such as the benefits of using larger batch sizes and deduplicated data, which can inform better model design and training practices.

**Weaknesses:**

This paper makes a valuable contribution to the field of factual knowledge acquisition during pretraining. While I'm not an expert in this area, I believe the paper is well-written and presents a strong argument. However, due to my limited experience, my confidence level is relatively low as I might have missed some key points.

**Questions:**

I only have several minor questions:
- Potential for Overlap with Factual Knowledge: While the proposed dataset is labeled as "fictional," it may not be entirely independent of existing factual knowledge. There's a possibility of overlap or influence between the fictional narratives and real-world information present in the training corpus. This could lead to interactions where the model's understanding of factual information might affect its interpretation of the fictional content, or vice versa.
- Potential for Bias and Contamination from ChatGPT: The use of ChatGPT for generating the dataset raises concerns about potential bias and contamination. Will these biases be reflected in the generated fictional narratives, potentially influencing the training of the model and leading to unintended consequences?

**Limitations:**

The authors included the limitation section in the Appendix.

---

> ### Author Rebuttal · Authors · 2024-08-06
>
> We sincerely thank reviewer fwEj for their time and effort in reviewing our work.
>
> We appreciate the reviewer's acknowledgment of the contributions of our work:
>
> - The creation of FICTIONAL KNOWLEDGE dataset for controlled experiments on factual knowledge acquisition dynamics in LLM pretraining
> - The comprehensive evaluation framework we developed, including metrics for assessing knowledge acquisition and retention
> - The practical insights provided by our research for improving LLM training practices through empirical analysis
> - The potential impact of our work on improving model reliability and performance, as well as its contribution to a more nuanced understanding of LLM pretraining
>
> Regarding the "poor" rating for contribution, we respectfully seek clarification. Given the reviewer's positive comments on the novelty of our dataset, evaluation framework, and insights, we believe that our work makes a significant contribution to the field. Any specific feedback will be greatly appreciated.
>
> Although the reviewer did not provide specific weaknesses or questions, we are open to further discussion on our work. We have addressed concerns and questions from other reviewers, and we would be happy to discuss these if it can be helpful for a more comprehensive evaluation of our paper.
>
> We thank the reviewer again for their time and valuable feedback.

---

> > ### Comment · Reviewer_yByP · 2024-08-09
> >
> > Thanks for the response. I want to apologize that the rating "poor" is simply a mistake. I changed it to "good" based on my overall assessment. Meanwhile, I have two specific questions for this paper. It would be appreciated if the authors could answer them. In general, I would like to keep my rating positive for this paper.

---

> > > ### Author Response · Authors · 2024-08-12
> > >
> > > Thank you for your clarification regarding the rating. We greatly appreciate you taking the time to correct this and for maintaining a positive assessment of our paper.
> > >
> > > ## Potential for Overlap with Factual Knowledge
> > >
> > > Although we strived to mitigate the interferences caused by the overlap between injected knowledge and pretraining corpus by designing fictional knowledge, we agree that this cannot completely erase such overlap. There is a trade-off between the designed knowledge being ‘fictitious’ to minimize overlap between knowledge in a pretraining corpus, and still being ‘realistic’ not to invoke a significant distribution shift, which can cause our investigation to deviate from the true dynamics of acquiring factual knowledge contained in the pretraining corpus.
> > >
> > > We conducted an additional analysis to understand the effect of such overlap on measured metrics. Specifically, we first estimated the overlap by examining the distribution of average BM25 ($k_1$=1.5, $b$=0.75) scores between our injected knowledge (used for duplication scenario) and 512,000 passages (about 1B tokens) extracted from the Dolma corpus used for our main experiments. Next, we took the top-10 and bottom-10 entries based on the average BM25 scores, and measured average effectivity ($\mathcal{E}(q,i)$) and decay constant (a) for each group, with experimental data from the OLMo-7B-mid checkpoint and semantic probes:
> > >
> > > | Avg. BM25 Score | Avg. $\mathcal{E}(q,i)$ | a    |
> > > |-----------------|-------------------------|------|
> > > | Top-10          | 0.47                    | 0.33 |
> > > | Bottom-10       | 0.41                    | 0.19 |
> > >
> > > This suggests that high overlap between given factual knowledge and pretraining data might lead to higher effectivity, but at the same time make forgetting faster, which is quite an interesting result. However, we note that this statement is inconclusive, as this is a case study with only a small set of samples and we leave a more thorough investigation on this topic for future work.
> > >
> > > Still, we believe these effects are mitigated in our analysis because: (1) we averaged all metrics across the entire dataset, and (2) our primary focus is on relative changes in log probability. Thus, we don't anticipate a significant impact on our core findings and interpretations.
> > >
> > > ## Potential for Bias and Contamination from ChatGPT
> > >
> > > Great question! As you pointed out, we observed that ChatGPT prefers to use several entity names (especially the names of humans) during dataset construction, which can lead to unintended consequences. To minimize the impact on training dynamics, we manually modified such cases, ensuring there are no overlapping fictional named entities between different injected knowledge instances. Second, to avoid a significant distribution shift during knowledge injection, we replaced only part of the original batch data with the injected knowledge, which takes up less than 10% of the batch tokens.

---

> > > > ### Comment · Reviewer_yByP · 2024-08-14
> > > >
> > > > Thanks for the clarifications. They certainly help to addressed some of my concerns. I would like to keep my rating positive for this paper.

---

### Official Review · Reviewer_ewnz · 2024-07-13

**Soundness:** 4
**Presentation:** 4
**Contribution:** 4
**Rating:** 8
**Confidence:** 4

**Summary:**

This study presents a comprehensive empirical analysis of how large language models (LLMs) acquire factual information during pre-training. The researchers used a novel and straightforward method: they introduced new fictional information into the training corpus and reran the pre-training process to observe whether the model could successfully recall this fictional information. The study focuses on four key factors: (1) the form of the injected factual information; (2) the timing of when the information is provided to the model; (3) the model size; and (4) the training batch size.
Their experiment confirms widely held intuitions about the process of factual knowledge acquisition and recall. However, it also uncovers surprising, counterintuitive results that challenge our previous understanding of LLMs.

**Strengths:**

I commend the authors for their excellent writing and impressive experimental setup. Conducting experiments on such a large scale is truly remarkable, and the findings provide valuable insights for future researchers on how factual information is acquired by LLMs during the pre-training process.

**Weaknesses:**

### Traces of Facts in Fiction

The fictional data used in the experiments is generated by ChatGPT. Although this data is fictional, it often incorporates factual information that the model has learned from its original training corpus.

For example, there is a piece of ficition information started with "The Southern Liberation Organization (SLO) is a cessationist, liberal democratic political party..." that talked about several fictional cessationist organisations. The evaluation input and targets (highlighted in bold) are:
1. Federalist Democracy of Korean Republic, abbreviated as FDKR, is another political alliance, which is working toward establishing autonomy for the southern provinces of **South Korea**.
2. Meanwhile, in North America, The Central Freedom Party (CFP) is striving for the independence of **the Central United States**.
3. An equally interesting political entity, the Western Autonomy League (WAL) advocates for the political autonomy of **Western Canada**.
4. In the Indian subcontinent, the Southern Freedom Group (SFG) seeks self-rule for **Southern India**.
5. In North Africa, the Saharan Independence Union (SIU) has been making efforts for the cause of separating Saharan region from the **African mainland**.

While these organizations are fictional, it will not be surprising that a language model can infer the expected fictional answers from context. For example, the Federalist Democracy of the Korean Republic is likely based in South Korea. Similarly, given the context of the Indian subcontinent, it is evident that the **Southern** Freedom Group supports the secession of Southern India. In summary, I believe that there is room for refinement.

### Release of Checkpoints

I appreciate that the authors have provided the code and data used in their experiments as supplementary documents. However, it remains unclear if the checkpoints of their model with injected information will be released. As mentioned in Appendix D, each experiment requires approximately three days of training using eight 80GB A100 GPUs. This level of computational resource is not accessible to many researchers. The checkpoints can be very helpful for researchers working in similar areas.

### The Mechanism of Knowledge Acquisition and Recall

There is a large body of work studying the process of which factual information is recalled from the model. Just to name a few:
- Geva et al., 2021. Transformer Feed-Forward Layers Are Key-Value Memories.
- Geva et al., 2023. Dissecting Recall of Factual Associations in Auto-Regressive Language Models
- Dai et al., 2022. Knowledge Neurons in Pretrained Transformers.
It would be interesting to see the connection and a discussion between this work and the mechanism research.

**Questions:**

1. Will you release the checkpoints with the injected fictional knowledge?

Please refer to the weakness section.

**Limitations:**

The authors have adequately addressed the limitations.

---

> ### Author Rebuttal · Authors · 2024-08-06
>
> We sincerely thank reviewer ewnz for their positive and thoughtful feedback. We appreciate the recognition of our contributions, particularly our insights on factual knowledge acquisition dynamics in LLM pretraining, as well as the commendation on our writing clarity and experimental design.
>
> We would like to address each point in order:
>
> ### **Traces of Facts in Fiction**
>
> We appreciate the reviewer’s thorough investigation of our FICTIONAL KNOWLEDGE dataset! This is a good point, and we acknowledge the trade-off between the designed knowledge being ‘fictitious’ to minimize overlap between knowledge in a pretraining corpus, and still being ‘realistic’ not to invoke a significant distribution shift, which can cause our investigation to deviate from the true dynamics of acquiring factual knowledge contained in the pretraining corpus. We agree there is room for improvement in controlling this trade-off during dataset construction.
> We believe these 'traces of facts' primarily affect the initial log probability the model assigns to each probe's target span before encountering the knowledge, while it may also influence the forgetting dynamics. However, we expect these individual effects to be mitigated in our analysis because: (1) we averaged all metrics across the entire dataset, and (2) our primary focus is on relative changes in log probability. Therefore, we don't anticipate a significant impact on our core findings and interpretations.
>
> ### **Release of Checkpoints**
>
> Unfortunately, we could not store the intermediate model checkpoints due to storage constraints. Instead, we logged the model's logits for every instance in our dataset at each training step. We understand the computational challenges in reproducing our experiments and will release these log files to facilitate further analysis and ensure transparency.
>
> ### **The Mechanism of Knowledge Acquisition and Recall**
>
> Thank you for highlighting recent works on knowledge acquisition and recall. We will incorporate mentions and citations to these works in the main text. We believe that combining the spatial aspects of locating factual knowledge within model parameters with our temporal approach presents an exciting direction for future research.
>
> We again thank the reviewer for their time and valuable feedback.

---

> > ### Comment · Reviewer_ewnz · 2024-08-12
> > **Acknowledgement to the rebuttal**
> >
> > Thank you for responding to my comments. I maintain my positive review for the paper.

---

### Official Review · Reviewer_ZYtT · 2024-07-13

**Soundness:** 4
**Presentation:** 4
**Contribution:** 4
**Rating:** 10
**Confidence:** 3

**Summary:**

The author's present a dataset and method to measure factual knowledge acquisition for LLM pretraining.

The author's conduct thorough investigations and effects (invented) factual knowledge during LLM pretraining and discover empirical trends and evidence for observations that have been recently observed (but not understood) in the research community.

**Strengths:**

Strengths:
1. Excellent method contribution to study pretraining in LLMs
2. Excellent analysis of performance and effects on performance for LLMs during pre-training

**Weaknesses:**

Weakness:
1. It would be interesting to understand how much overlap of the proposed dataset there is with the pretraining dataset.
2. It would be interesting to understand if forgetting is more or less pronounced when follow-on tokens are similar or different to the injected knowledge.
3. Log-prob metric is clear, the others are a bit vague.

**Questions:**

1. How could you actually measure the "forgetting period" for long-tail knowledge?

**Limitations:**

Yes.

---

> ### Author Rebuttal · Authors · 2024-08-06
>
> We sincerely thank reviewer ZYtT for acknowledging our contribution to studying the pretraining dynamics in LLMs, and commendation to our method and analysis regarding this topic. Also, we appreciate the reviewer’s effort to improve our work.
>
> We would like to discuss each point in order:
>
> > It would be interesting to understand how much overlap of the proposed dataset there is with the pretraining dataset.
>
> This is a good point! Understanding the overlap between our FICTIONAL KNOWLEDGE dataset and the pretraining corpus, and its impact on acquisition dynamics, would indeed provide valuable insights. While we haven't conducted a quantitative evaluation of how lexical and semantic overlap affects the acquisition of each knowledge piece, we hypothesize that such overlap primarily influences the initial log probability the model assigns to each probe's target span before encountering the knowledge. As the reviewer noted in their second point, the overlap between given knowledge and follow-up tokens may also affect forgetting dynamics. However, we believe these individual effects are mitigated in our analysis because: (1) we averaged all metrics across the entire dataset, and (2) our primary focus is on relative changes in log probability. Thus, we don't anticipate a significant impact on our core findings and interpretations.
>
> > It would be interesting to understand if forgetting is more or less pronounced when follow-on tokens are similar or different to the injected knowledge.
>
> We agree this is an intriguing direction! Analyzing the influence of given knowledge's similarity to follow-on tokens on the forgetting dynamics will provide valuable insights.
>
> > Log-prob metric is clear, the others are a bit vague.
>
> We acknowledge that some of our metrics may not be immediately intuitive. However, we would like to note that our primary focus on designing the metrics was making them well-defined and easily adoptable in future works on analyzing fine-grained pretraining dynamics with different experimental setups. This is of particular importance as this is one of the first works to study this, although it may come at the slight cost of clarity. We will strive to improve the clarity of our metrics in the revised version, ensuring readers can easily grasp the meaning of each metric while maintaining their adaptability.
>
> > How could you actually measure the "forgetting period" for long-tail knowledge?
>
> As discussed in the footnote in Section 4.4 (L270), we didn't explicitly measure the forgetting period (which we interpret as the learnability threshold) for long-tail knowledge as our concept of learnability threshold is derived from our simulated setup. Therefore, the theoretical period we discuss may not exactly match the estimated x-intercept in Figure 5. However, we believe our results provide a reasonable estimate for relative comparisons of learnability thresholds.
>
> We thank the reviewer again for their time and valuable feedback.

---

> ### Comment · Reviewer_ZYtT · 2024-08-08
> **Further questions:**
>
> Thank you for taking time to respond to the weaknesses and questions.
>
> Your rebuttal, while generally clear, throws up a few more questions due to quite surface level answers.
>
> > (1) we averaged all metrics across the entire dataset, and
>
> Averaging does not answer the question of how much overlap there is (only if it was a `perfectly' balanced dataset). Could you provide an analysis or study of overlap between pre-training data (PTD) and Fictional Knowledge (FK)? Could you then provide additional results on how metrics differ for these categories?
>
> > (2) our primary focus is on relative changes in log probability. Thus, we don't anticipate a significant impact on our core findings and interpretations.
>
> Similarly, here it would be very interesting to see how relative log probability changes behave under varying overlapping conditions. As other reviewers noted some information could be easily inferred by the LLM in general and therefore the change might be small for such knowledge (and the forgetting rate as well).
>
> >  focus on designing the metrics was making them well-defined and easily adoptable in future works on analyzing fine-grained pretraining dynamics with different experimental setups
>
> This needs further clarification and concrete examples and demonstrations.
>
> >  We will strive to improve the clarity of our metrics in the revised version
>
> Could you give an attempt already now?
>
> Once again thank you for submitting the rebuttal, please elaborate on your answers and add more details. At this stage some things have become less clear.

---

> > ### Author Response · Authors · 2024-08-09
> > **Answer to the Follow-up Questions**
> >
> > > Could you provide an analysis or study of overlap between pre-training data (PTD) and Fictional Knowledge (FK)?
> >
> > Although we could not perform the analysis on the whole Dolma corpus, we estimated the overlap by examining the distribution of average BM25 ($k_1$=1.5, $b$=0.75) scores between our injected knowledge (used for duplication scenario) and 512,000 sequences (about 1B tokens) extracted from the Dolma corpus used for our main experiments. The analysis shows that the distribution can be fitted to a normal distribution with $\mu$=30.6 and $\sigma$=4.6, which we can say is a concentrated distribution. (Similarly, the distribution of averaged top-10 BM25 scores for each injected knowledge can be fitted to a normal distribution with $\mu$=118 and $\sigma$=16.2)
> > > Could you then provide additional results on how metrics differ for these categories?
> >
> > > it would be very interesting to see how relative log probability changes behave under varying overlapping conditions.
> >
> > To investigate these, we took the top-10 and bottom-10 entries based on the average BM25 scores, and measured average effectivity ($\mathcal{E}(q,i)$) and decay constant (a) for each group, with experimental data from the OLMo-7B-mid checkpoint and semantic probes:
> >
> > | Avg. BM25 Score | Avg. $\mathcal{E}(q,i)$ | a    |
> > |-----------------|-------------------------|------|
> > | Top-10          | 0.47                    | 0.33 |
> > | Bottom-10       | 0.41                    | 0.19 |
> >
> > This result suggests that high overlap between given FK and PTD might lead to higher effectivity. But at the same time, this might make forgetting faster, which is quite an interesting result. However, we highlight that this statement is inconclusive, as this is a small-scale case study and the result may not be statistically significant. We leave a more thorough investigation on this topic for future work, as this will require a more focused experimental design and another pretraining experiment.
> >
> > > This needs further clarification and concrete examples and demonstrations.
> >
> > I will provide an example regarding the design of effectivity. Suppose we are trying to measure ‘how much log probability is increased due to the training on a given FK?’ without considering i (which means the i-th encounter of the FK we evaluate) for simplicity and clarity. Then, we have two options:
> > 1. Measuring the log probability increases before and after the whole training session: As demonstrated in Figure 2, the loss of log probability due to forgetting starts immediately after the injection. Therefore, the log probability increases driven by FK injection will be underestimated, unless we inject FK only once or multiple times in a very short interval. Moreover, the amount of such underestimation will depend on the injection intervals, restricting the direct comparison of the measured values between different training setups.
> > 2. Measuring the immediate log prob increases any time the model encounters the FK: As shown in Appendix H, the amount of the immediate increase is affected by the number of previous encounters, and it is especially high when the model has not encountered the knowledge before. Therefore, this option will compromise the interchangeability of effectivity between works experimenting with different numbers of injections.
> >
> > There were similar considerations for the design of retainability (how much log probability decayed due to forgetting). In short, we measured the fraction of the log probability the model retained t steps after the final LAM, as the rate of forgetting may depend on the number of FK injections.
> >
> > > Could you give an attempt already now?
> >
> > We will make the following modifications to improve clarity and readability:
> > - First, as Reviewer 1qMb pointed out, we will change the symbol to represent effectivity (e.g., *Eff*), as the previous one ($\mathcal{E}$) can be misinterpreted as an error.
> > - Second, we will simplify the notations without changing their meaning. For example, we can simplify the notation of a model’s log probability on probe $q$ at timestep $t$ from $\ell(q,\theta_t)$ to $\ell(q,t)$.
> > - Third, we will elaborate on the intuitions behind the metric design and its meaning in more detail in the main text.
> >
> > We hope this clarifies your follow-up questions. Thank you for taking the time to address this matter.

---

> > > ### Comment · Reviewer_ZYtT · 2024-08-11
> > >
> > > Thank you for your clarifications and taking the time to run additional experiments.
> > >
> > > —
> > > A small follow-up in your BM-25 experiment. A very interesting observation on retainability. What do you think might be the reason for a higher a for more overlap?

---

> > > > ### Author Response · Authors · 2024-08-12
> > > >
> > > > We appreciate the reviewer's insight regarding our work, which led us to a very interesting analysis!
> > > >
> > > > Our guess about the reason for a higher decay constant for the instances with high overlap is based on the nature of softmax computation. When the model encounters a context that does not entail the FK of interest but has an overlap, the model will be driven to increase the logit of the context, which essentially reduces the log probability of the other options. We guess that this occurs more frequently for FK that has a high overlap with the PTD, which may boost forgetting.

---

> > > > > ### Comment · Reviewer_ZYtT · 2024-08-12
> > > > >
> > > > > Thank you. Indeed, these are very interesting insights.

---

### Author Rebuttal · Authors · 2024-08-06

We sincerely thank all reviewers for their commitment to thoroughly reviewing our work. We greatly appreciate the reviewers’ recognition of this work’s contributions:

- **The development of a novel dataset (FICTIONAL KNOWLEDGE), experimental methods, and evaluation metrics to study how knowledge is acquired during LLM pretraining (Reviewer ZYtT, ewnz, yByP, fwEj, and 1qMb)**
- **The insights provided by our comprehensive analysis of LLM pretraining, confirming several recently observed yet underexplained behaviors of LLMs and challenging our previous understanding of LLMs (Reviewer ZYtT, ewnz, yByP, fwEj, and 1qMb)**
- **The potential impact of our work on improving the reliability and performance of LLMs (Reviewer ewnz, yByP, and fwEj)**

Despite this work's contributions, we acknowledge several limitations. We address the major points below, focusing on the most significant issues due to space constraints:

**Limited evaluation of varying learning rates and batch sizes (Reviewer fwEj and 1qMb)**

We acknowledge this limitation, as noted in our Limitations section (L504-506). Given the significant computational resources required for each pretraining experiment (L690-691), exploring a wide range of learning rates and batch sizes was not feasible within our current academic constraints. However, we emphasize that our experiments used the learning rate and batch size used for actual OLMo pretraining, ensuring our results have significant implications for recent LLMs in practice.

We recognize the importance of this aspect and aim to address it in future work if additional computational resources become available.

**Clarity of the evaluation metrics (Reviewer ZYtT and 1qMb)**

We acknowledge that some of our metrics may not be immediately intuitive. However, we would like to note that our primary focus on designing the metrics was making them easily adoptable in future works on analyzing fine-grained pretraining dynamics with different experimental setups. This is of particular importance as this is one of the first works to study this, although it may come at the slight cost of clarity. We will strive to improve the clarity of our metrics in the revised version, ensuring readers can easily grasp the meaning of each metric while maintaining their adaptability.

**Overlap between real knowledge and fictional knowledge (Reviewer ZYtT and ewnz)**

We acknowledge the trade-off between the designed knowledge being ‘fictitious’ to minimize overlap between knowledge in a pretraining corpus, and being ‘realistic’ not to invoke a significant distribution shift. We hypothesize that such overlap primarily influences the initial log probability assigned to each probe's target span before the knowledge injection. Although the overlap may also affect the forgetting dynamics, we believe that these individual effects are mitigated in our analysis because: (1) we averaged all metrics across the entire dataset, and (2) our primary focus is on relative changes in log probability. Therefore, we don't anticipate a significant impact on our core findings and interpretations.

While we couldn't address every detail here, we've carefully considered all feedback and are committed to addressing both major and minor points in our revision.

We sincerely thank all reviewers again for the constructive feedback.

---

### Decision · Program_Chairs · 2024-09-25

**Decision:**

Accept (poster)

**Comment:**

This paper studies how LLMs acquire factual knowledge during pretraining. This paper collects the Fictional Knowledge dataset, containing passages that describe fictional yet realistic entities. These passages are injected into the sequence in a pretraining batch. The experiments reveal many insightful observations.

The reviewers commended the experimental setup, the writing, and considered that the findings to be valuable contributions to the field.

The concerns about the potential overlap between the generated data and the pretraining corpora were discussed, and I don’t think this weakens the analysis of the paper. The concern about bias is addressed well.

One reviewer gave a negative score, and most of the concerns are addressed. The main concern, the interpretability of the log probability itself, is addressed well by the followup response of the author.